# Tyrosine phosphorylation of CARM1 promotes its enzymatic activity and alters its target specificity

Hidehiro Itonaga [1], Adnan K. Mookhtiar[1], Sarah M. Greenblatt[1,2], Fan Liu [1,3], Concepcion Martinez[1], Daniel Bilbao [1], Masai Rains[1], Pierre-Jacques Hamard [1,4], Jun Sun[1,5], Afoma C. Umeano[6], Stephanie Duffort[1], Chuan Chen [1], Na Man[1], Gloria Mas [1], Luca Tottone [1], Tulasigeri Totiger[1], Terrence Bradley[7], Justin Taylor [1], Stephan Schürer [6] & Stephen D. Nimer [1,3,5] ✉

An important epigenetic component of tyrosine kinase signaling is the phosphorylation of histones, and epigenetic readers, writers, and erasers. Phosphorylation of protein arginine methyltransferases (PRMTs), have been shown to enhance and impair their enzymatic activity. In this study, we show that the hyperactivation of Janus kinase 2 (JAK2) by the V617F mutation phosphorylates tyrosine residues (Y149 and Y334) in coactivator-associated arginine methyltransferase 1 (CARM1), an important target in hematologic malignancies, increasing its methyltransferase activity and altering its target specificity. While non-phosphorylatable CARM1 methylates some established substrates (e.g. BAF155 and PABP1), only phospho-CARM1 methylates the RUNX1 transcription factor, on R223 and R319. Furthermore, cells expressing non-phosphorylatable CARM1 have impaired cell-cycle progression and increased apoptosis, compared to cells expressing phosphorylatable, wild-type CARM1, with reduced expression of genes associated with G2/M cell cycle progression and anti-apoptosis. The presence of the JAK2-V617F mutant kinase renders acute myeloid leukemia (AML) cells less sensitive to CARM1 inhibition, and we show that the dual targeting of JAK2 and CARM1 is more effective than monotherapy in AML cells expressing phospho-CARM1. Thus, the phosphorylation of CARM1 by hyperactivated JAK2 regulates its methyltransferase activity, helps select its substrates, and is required for the maximal proliferation of malignant myeloid cells.

Protein arginine methylation is an essential protein post-translational modification, with ~7% of arginine residues being methylated, which is comparable to the 9% of serine residues that are phosphorylated and the 7% of lysine residues that are ubiquitinated[1]. Protein arginine methyltransferases (PRMTs) catalyze monomethylation, asymmetric dimethylation, or symmetric dimethylation reactions on arginine residues[2], and are classified as class I (asymmetric dimethyl arginine; ADMA), class II (symmetric dimethyl arginine; SDMA) and class III methyltransferases (monomethyl arginine; MMA)[2–4]. PRMTs are ubiquitously expressed and they regulate multiple cellular processes, including transcription, RNA splicing, DNA replication, DNA repair, protein translation, and cellular

---

metabolism, thereby affecting cell growth, proliferation, and differentiation[5-11].

Coactivator-associated arginine methyltransferase 1 (CARM1), also known as PRMT4, was originally identified as a coactivator for steroid hormone receptors[12]. CARM1 is a type I protein arginine methyltransferase (PRMT), that catalyzes the asymmetric dimethylation of arginine residues in histones, such as H3R17 and H3R26, which are thought to promote transcription[13,14]. Furthermore, CARM1 also catalyzes the ADMA of non-histone substrates, including transcription factors like RUNX1[15], histone acetyltransferases (e.g. p300 and CBP)[16-18], the steroid receptor co-activator AIB1[19,20], RNA binding proteins (e.g. PABP1)[21], RNA splicing factors (e.g. SAP49, SmB, and U1C)[22], and components of the SWI/SNF chromatin remodeling complex (e.g. BAF155)[16,23].

We previously demonstrated that CARM1 blocks the myeloid differentiation of normal hematopoietic stem/progenitor cells (HSPCs) by promoting the assembly of a repressive RUNX1 complex[15], and that *Carm1* knockout in adult mouse HSPCs prevents the development of acute myeloid leukemia (AML), driven by either the *AML1::ETO* or *MLL::AF9* oncogenes, but only modestly decreases long-term hematopoietic stem cell (HSC) numbers[24]. *Carm1* knockout also abrogated the maintenance of AML, suggesting that CARM1 inhibition could have therapeutic efficacy in AML[24]. CARM1 is overexpressed in a variety of cancers, including breast, lung, colorectal, liver, and prostate cancer[25-30] and because it drives key oncogenic processes, it could be a therapeutic target in these diseases as well.

While evaluating the therapeutic relevance of CARM1 inhibition, we observed the differential sensitivity of AML cell lines and noticed that JAK2-V617F mutation-positive myeloid leukemia cells showed a lower sensitivity to CARM1-inhibition[24]. Several post-translational modifications (PTMs) of CARM1 have been reported, including its serine phosphorylation and arginine auto-methylation[31-34], and we have been defining how signaling pathways, such as the Janus-activated tyrosine kinase (JAK) family (JAK1, JAK2, JAK3, and TYK2) which are activated by cytokine receptors and other cell surface receptors, affect the activity of epigenetic modifiers[35-37].

Activated JAK2 signals through downstream effectors including the signal transducers and activators of transcription (STAT) transcription factors (TFs), and the Ras-mitogen-activated protein kinase (MAPK) and phosphoinositide 3-kinase (PI3K) pathways, which regulate hematopoietic cell differentiation, cell proliferation, and apoptosis[36,38,39]. A single somatic mutation, V617F (exon 14) in the JH2 domain of JAK2, which disrupts the JH1-JH2 autoinhibitory interaction, leading to JAK2 hyperactivation[40-42], is a common hallmark of the *BCR::ABL1*-negative myeloproliferative neoplasms. It is found in >95% of polycythemia vera patients, with homozygous mutations found in patients with longstanding disease. It is also found in ~30-50% of patients with essential thrombocythemia and primary myelofibrosis, and in ~1-4% of adult AML and myelodysplastic syndrome patients[42-45].

JAK2 signaling can alter chromatin structure by directly phosphorylating epigenetic proteins, such as TET2 and histone H3.1[46,47]. We have reported that JAK2 is both cytoplasmic and nuclear, and that it phosphorylates PRMT5 resulting in the downregulation of its methyltransferase activity on histones[48], we hypothesize that tyrosine phosphorylation by JAK2 could also alter the function of CARM1.

In this study, we determine that when the JAK2-V617F kinase is itself phosphorylated (and activated), it phosphorylates CARM1 on tyrosine-149 and −334 (which are located within the CARM1 catalytic domain), promoting its methyltransferase activity and altering its nuclear localization. Based on multiple in vitro and in vivo studies, we also find that CARM1 phosphorylation alters its substrate specificity and its effects on CARM1 target gene selection. We see important biological differences in the ability of phosphorylatable *vs.* non-phosphorylatable CARM1 to support leukemia cell proliferation and demonstrate the therapeutic relevance of targeting both JAK2-V617F

and CARM1 in JAK2-V617F-positive cells. These results provide further insights into the regulation of chromatin structure by tyrosine kinases and the pathogenesis and treatment of myeloid neoplasms.

## Results

### Identification of tyrosine residues in CARM1 phosphorylated by JAK2

To determine whether CARM1 is a direct substrate of JAK2, and identify potential CARM1 tyrosine phosphorylation sites, we performed cell-free in vitro kinase assays using GST-tagged CARM1 protein as the substrate for a recombinant active form of JAK2 kinase or recombinant PAK1 (a known JAK2 substrate) as a positive control (Fig. 1A). The JAK2-dependent tyrosine phosphorylation of GST-tagged CARM1 was readily identified (lanes 4 and 5), and incorporation of a JAK1/JAK2 inhibitor (ruxolitinib [RUX]) in the kinase assay completely abrogated CARM1 phosphorylation (lane 7). We next identified tyrosine-149 (Y149) and tyrosine-334 (Y334) as the sites of JAK2 phosphorylation by subjecting the in vitro phosphorylated, recombinant GST-tagged CARM1 protein to mass spectrometry analysis (Fig. 1B–E and Supplementary Fig. 1A). These tyrosine residues are located within the core catalytic domain of CARM1 (Fig. 1F), and are highly conserved in CARM1 proteins from African clawed frog to humans (Supplementary Fig. 1B, C).

### CARM1 phosphorylation is mediated by JAK2 in myeloid leukemia cells

We generated two phospho-tyrosine-specific rabbit polyclonal antibodies, against either Y149 phosphorylated or Y334 phosphorylated CARM1 (Supplementary Fig. 2A); these antibodies recognize in vitro phosphorylated recombinant GST-tagged CARM1 protein, but not the unphosphorylated protein, in a dose-dependent manner (Supplementary Fig. 2B, C). Furthermore, these antibodies recognize MYC-tagged wild-type (WT) CARM1, but not the non-phosphorylatable MYC-tagged CARM1 mutant proteins (that contain Y149F, Y334F, or Y149F/Y334F amino acid substitutions), based on the in vitro assays (with or without active JAK2 kinase) using the immunoprecipitants from K562 cell lysates (Supplementary Fig. 2D).

We then compared the level of CARM1 protein expression and CARM1-Y149 and -Y334 phosphorylation in 14 myeloid leukemia cell lines, using normal human CD34+ cord blood (CB) cells as control. As previously described[15,24], we found that CARM1 expression levels are higher in nearly all of these myeloid leukemia cell lines, compared to the normal CD34+ CB cells (Fig. 2A). HEL cells, that express JAK2-V617F, showed the highest level of phosphorylated CARM1-Y149 and -Y334 (lane 2), while SET2 cells, which also express JAK2-V617F, showed abundant CARM1 protein but a lower relative amount of phosphorylated CARM1 protein (lane 3) than the HEL cells. K562 cells (lane 8) had abundant CARM1, but less phosphorylated CARM1. Treating HEL cells with the JAK2 inhibitor (RUX, 500 nM or NVP-BSK856[49]) decreased the phosphorylation of CARM1-Y149 and -Y334, and STAT5 (Fig. 2B, and Supplementary Fig. 2E, F), confirming that JAK2 is one of the relevant kinases that phosphorylate CARM1.

### Activated JAK2 binds to and phosphorylates CARM1

To determine whether JAK2 protein interacts with CARM1 in vivo, we transduced HEL, SET2, and K562 cells with an HA-tagged CARM1 construct, immunoprecipitated using an anti-HA antibody, followed by immunoblotting with an anti-JAK2 antibody. We were able to detect the direct binding of JAK2 to CARM1 in HEL cells and to a lesser extent in SET2 cells, but weakly in K562 cells (Fig. 2C). To determine what accounts for these differences, we focused on the phosphorylation status of JAK2 in these cells, as HEL cells have a bi-allelic *JAK2*-V617F mutation, while SET2 cells have a mono-allelic *JAK2*-V617F mutation (Supplementary Fig. 2A). Biallelic mutations lead to a higher level of JAK2 auto-phosphorylation, leading us to hypothesize that the bi-allelic *JAK2*-V617F mutation in HEL cells triggers the higher level of

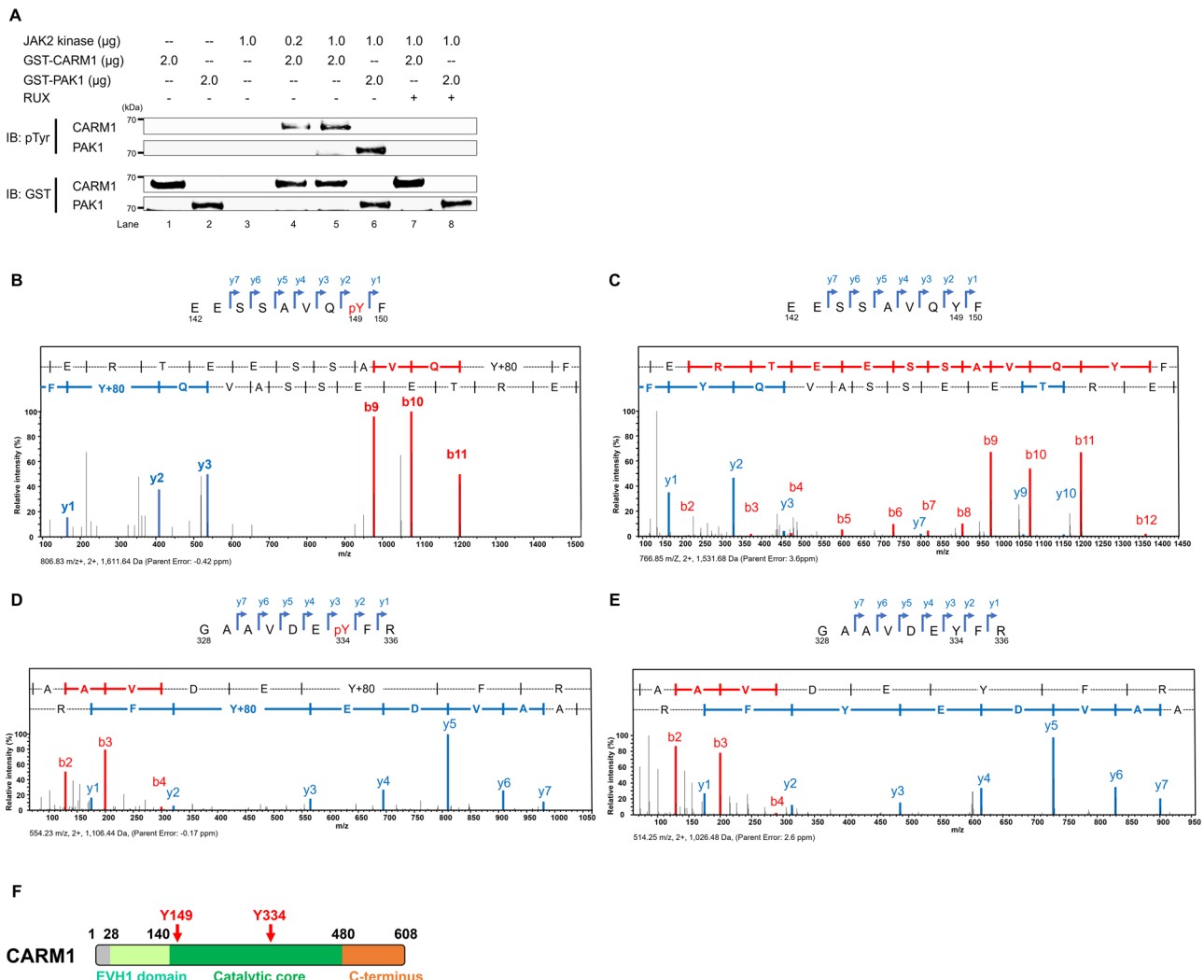

**Fig. 1 | JAK2 phosphorylates CARM1 in vitro. A** JAK2 phosphorylates CARM1 in an in vitro kinase assay, in which active JAK2 kinase and recombinant CARM1 proteins were used; PAK1 was used as a positive control. The amounts of protein in the reaction are indicated. The phosphorylation of CARM1 was completely abolished by the JAK1/JAK2 inhibitor (ruxolitinib, RUX; 500 nM) (lanes 7 and 8). The experiment was repeated independently twice. **B** Peptide fragments in the mass spectrometry analysis were generated from proteolytic cleavage of CARM1 following in vitro kinase assays in the presence of active JAK2 kinase. Tyrosine-149 (Y149) along with the series of y- and b-ions, including the phosphorylated residue, is shown as the phosphorylated peptide (EESSAVQpYF). **C** Peptide fragments in the mass spectrometry analysis were generated from proteolytic cleavage of CARM1 following in vitro kinase assays in the absence of active JAK2 kinase. The peptide fragment around Y149 residue (EESSAVQYF) is shown without phosphorylation of tyrosine, indicating no gain in molecular weight of 80 Da (i.e. the weight of PO$_4$). **D** Peptide fragments in the mass spectrometry analysis were generated from proteolytic cleavage of CARM1 following in vitro kinase assays in the presence of active JAK2 kinase. Tyrosine-334 (Y334) along with the series of y- and b-ions, including the phosphorylated residue, is shown as the phosphorylated peptide (GAAVDEpYFR). **(E)** Peptide fragments in the mass spectrometry analysis were generated from proteolytic cleavage of CARM1 following in vitro kinase assays in the absence of active JAK2 kinase. The peptide fragment around Y334 residue (GAAVDEYFR) is shown without phosphorylation of tyrosine, indicating no gain in molecular weight of 80 Da. **F** The regions containing amino acid residues Y149 and Y334 are located within the core catalytic domain (residue 140-480) of CARM1. Residues 28-140 in CARM1 are highly homologous to a family of *Drosophila*-enbaled/vasodilator-stimulated phosphoprotein homology 1 (EVH1) domains, which specifically bind to target proline-rich sequences with low affinity and high specificity.

CARM1 phosphorylation. First, we confirmed that HEL cells have the highest level of JAK2 phosphorylation among the various myeloid leukemia cell lines tested, including SET2 cells (Supplementary Fig. 3B). We also examined the UKE-1 cell line, derived from a myeloproliferative neoplasm patient that carries a bi-allelic *JAK2*-V617F mutation[50], and show that UKE-1 cells also have a high level of JAK2 auto-phosphorylation, as well as CARM1-Y149/Y334 phosphorylation (Fig. 2D). None of the other JAK family members (JAK1, JAK3 or TYK2) bound CARM1 in HEL cells, even when CARM1 was overexpressed (Supplementary Fig. 3C).

We next examined whether JAK2 auto-phosphorylation affected the binding of JAK2 to CARM1 or its kinase activity on CARM1, using an anti-HA antibody for the immunoprecipitation and an antibody against Y1007/Y1008 phosphorylated JAK2 for immunoblotting. Phosphorylated JAK2 bound CARM1 in HEL cells that overexpress HA-tagged CARM1 (Fig. 2E). Given that JAK1 and TYK2 activity can transphosphorylate JAK2[51], we generated JAK1 and TYK2 knockout (KO) HEL cells using the clustered regularly interspaced short palindromic repeat (CRISPR)/CRISPR-associated protein-9 (Cas9) nuclease system. KO of either JAK1 or TYK2 decreased the phosphorylation of JAK2 (by 18-28%) and CARM1 (by 38-55%) (Fig. 2F), but neither KO reduced the binding of JAK2 to CARM1 (Supplementary Fig. 4A). To confirm that the binding of JAK2 to CARM1 is JAK2-phosphorylation independent, we used a type II JAK2 inhibitor, CHZ868[52]. CHZ868 treatment abrogated JAK1-mediated JAK2-Y1007/Y1008 phosphorylation (Supplementary Fig. 4B), but it did not reduce the binding of JAK2 to CARM1.

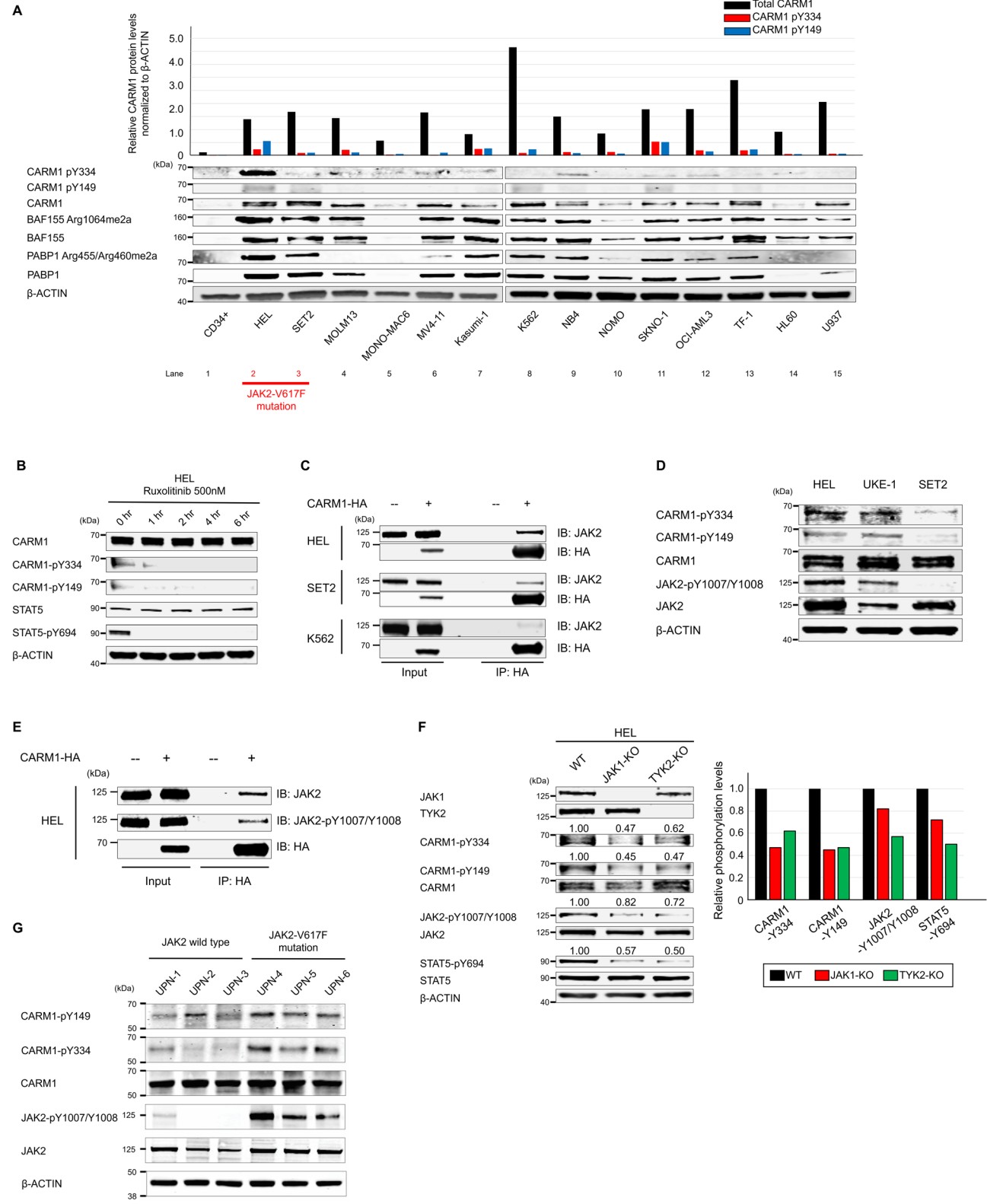

Taken together, these results indicate that JAK2 activation, through transphosphorylation by JAK1 and TYK2, enhances JAK2 tyrosine kinase activity against CARM1, but not via effects on the binding of JAK2 to CARM1.

To confirm the clinical relevance of CARM1 phosphorylation, we examined myeloid cells from patients with various myeloid malignancies. We purified mononuclear cells from patient bone marrow aspirates (Fig. 2G) and found that all three JAK2-V617F mutation-positive samples (Fig. 2G, lanes 4 to 6) had higher levels of both phospho-CARM1 and phospho-JAK2 than the JAK2-WT samples (lanes 1 to 3). We also observed greater phospho-CARM1 in the samples with more phospho-JAK2 (lane 4), than those with less phospho-JAK2 (lanes 5 and 6). The clinical information and gene mutational profile for each patient is summarized in Supplementary Table 2.

**Fig. 2 | JAK2-V617F promotes the tyrosine phosphorylation of CARM1 in myeloid leukemia cells. A** The expression of CARM1 protein and Y149/Y334 phosphorylated CARM1 protein was assessed in 14 myeloid leukemia cell lines and human CD34+ cord blood cells, by immunoblotting analysis. **B** Phosphorylation of Y149 and Y334 of CARM1 in HEL cells is abolished following treatment with the JAK1/JAK2 inhibitor (RUX at low concentration to avoid severe apoptosis; 500 nM), as is phosphorylation of tyrosine-694 in STAT5. **C** Immunoprecipitation was performed using HEL, SET2, and K562 cells that express HA-tagged CARM1, with an anti-HA antibody. Immunoblotting with anti-HA and anti-JAK2 antibodies revealed the interaction between JAK2 and CARM1. **D** HEL and UKE-1 cells harboring homozygous JAK2-V617F mutations had phosphorylated CARM1-Y149/Y334 and (auto) phosphorylated JAK2-Y1007/Y1008, while SET2 cells harboring heterozygous JAK2-V617F mutation did not. **E** Proteins were immunoprecipitated from HEL cell extracts that express HA-tagged CARM1, using an anti-HA antibody; immunoblotting was then performed using an anti-HA antibody, anti-phosphorylated JAK2 rabbit antibody, or anti-JAK2 mouse antibody. **F** Extracts from JAK1 or TYK2 knockout HEL cells were immunoblotted using phosphospecific anti-CARM1, JAK2, and STAT5 antibodies. **G** Phospho-CARM1 and phospho-JAK2 were evaluated in mononuclear cells isolated from patients with the following myeloid neoplasms: UPN-1 (unique patient number-1), chronic myeloid leukemia blast phase; UPN-2, acute myeloid leukemia with mutated *NPM1*; UPN-3, acute myeloid leukemia not otherwise specified; UPN-4, essential thrombocythemia; UPN-5, primary myelofibrosis; and UPN-6, polycythemia vera. All experiments were repeated at least two times independently. **A** n = 3, **B**–**G** *n* = 2.

## Tyrosine phosphorylation of CARM1 increases its methyltransferase activity

To better understand how the phosphorylation of CARM1 affects its substrate binding and methyltransferase activity, we analyzed published crystal structures of CARM1 and a known CARM1 substrate, PABP1[14,21]. CARM1-Y149 and -Y334 phosphorylation are predicted to increase the binding of CARM1 to unmethylated PABP1 (Figs. 3A, B). Y149 and Y334 phosphorylation could also affect the methyltransferase activity of CARM1, so we purified MYC-tagged CARM1 protein from 293 T cells transfected with WT or non-phosphorylatable mutant CARM1 (using the Y-to-F mutant proteins) and examined the ability of these mammalian cell-expressed CARM1 proteins to methylate histone H3.1 on arginine-17 (R17) in vitro and in vivo. The incubation of WT-CARM1 with S-[methyl-¹⁴C]-adenosyl-methionine and recombinant histone H3.1 showed significant methyltransferase activity in an in vitro methylation assay. However, none of the non-phosphorylatable CARM1 proteins had methyltransferase activity against histone H3.1 (Fig. 3C). Next, we examined whether incubating bacterially expressed CARM1 with an active JAK2 kinase affected its ability to methylate histone H3.1 in vitro. First, we confirmed the in vitro phosphorylation of CARM1, and then incubated histone H3.1 with S-[methyl-¹⁴C]-adenosyl-methionine and phosphorylated or non-phosphorylated CARM1. The addition of JAK2 kinase clearly increased CARM1 methyltransferase activity on histone H3.1 (Fig. 3D, lane 9 *vs.* lane 7), and the mass spectrometry analysis further confirmed the increased methylation of histone H3R17 by phosphorylated CARM1, compared to un-phosphorylated CARM1 (Supplementary Fig. 5).

## The Y149F mutation in CARM1 impairs its in vivo dimerization

CARM1 dimer formation involves interactions between the so-called dimerization arm (residues 300-338) and helices αX, αY, αZ, αA, and αB (residues 144-232)[53]. The sites of CARM1 phosphorylation (Y149 and Y334) are located within regions involved in CARM1 dimerization (Fig. 3E), and we demonstrated that the JAK1/JAK2 inhibitor (RUX, 10 μM) decreased CARM1 dimerization in 293 T cells, using HA-tagged WT-CARM1 and MYC-tagged CARM1 constructs for transfection study (Supplementary Fig. 6A). We co-transfected HA-tagged WT-CARM1 and MYC-tagged CARM1 constructs into 293 T cells overexpressing either the WT-JAK2 or JAK2-V617F mutant and saw that expression of the JAK2-V617F mutant kinase increased dimerization greater than expression of WT-JAK2 (Supplementary Fig. 6B). To investigate whether individual tyrosine phosphorylation sites in CARM1 affected dimerization, we co-transfected HA-tagged WT-CARM1 and MYC-tagged CARM1 (WT and Y-to-F mutated) constructs into 293 T cells, immunoprecipitated with anti-HA antibodies, and probed with anti-MYC antibodies. MYC-tagged CARM1-WT and -Y334F mutant CARM1 were readily detectable in the HA immunoprecipitates (Fig. 3F, lanes 7 and 8). However, the Y149F mutant and the Y149F/Y334F double mutant CARM1 proteins showed reduced binding to HA-tagged WT, compared to MYC-tagged WT CARM1 (Fig. 3F, lane 9 and 10 *vs.* lane 7). Thus, it appears that phosphorylation of Y149 in CARM1 (but not Y334) promotes its dimerization.

## CARM1 tyrosine phosphorylation promotes its nuclear localization

We investigated whether phosphorylation affects the subcellular localization of CARM1 using an anti-CARM1 antibody. While CARM1 localizes mainly in the cytoplasm of HEL, SET2, and K562 cells, some CARM1 is found in the nuclear soluble fraction (Supplementary Fig. 6C). Using CARM1 phospho-specific antibodies, and HEL and UKE-1 cells, we found a significantly higher proportion of Y149 and Y334 phosphorylated CARM1 in the nucleus, and the chromatin fraction, of both HEL cells and UKE-1 cells (Fig. 3G). We also assessed the subcellular localization of CARM1 in HEL cells following treatment with RUX (or DMSO); RUX treatment decreased the proportion of CARM1 in the nucleus and chromatin fraction (Supplementary Fig. 6D), indicating that the tyrosine phosphorylation of CARM1 contributes to its nuclear and chromatin localization.

## Identification of phospho-CARM1 interacting RUNX1

To capture the proteins which are associated with phosphorylated CARM1, we engineered HEL cells (with abundant phosphorylated CARM1) and K562 cells (with primarily non-phosphorylated CARM1) to express the proximity-dependent biotin identification (BioID) system (Supplementary Fig. 7A and B). Using shotgun mass spectrometry, we identified 128 and 60 proteins that significantly interact with CARM1 in HEL and K562 cells, respectively (supplementary data, and Supplementary Fig. 7C, D). We identified many previously known CARM1-associated proteins or substrates, including NUDT4, ADAR, RUNX1, SON, and CARM1 in HEL cells but not in K562 cells[14,15] (Fig. 4A), and confirmed the high intensity of RUNX1 fragments interacting with CARM1-BirA* fusion in HEL cells, using target mass spectrometry (Supplementary Fig. 7E).

Having identified arginine-223 (R223) in RUNX1 as a site of CARM1 asymmetric dimethylation[15], we examined whether the tyrosine phosphorylation of CARM1 affects its ability to bind or methylate RUNX1. To examine the binding of CARM1 to RUNX1, we generated HEL cells overexpressing MYC-tagged CARM1 (WT, Y334F, Y149F, and Y149F/Y334F mutants), and immunoprecipitated the WT or non-phosphorylatable mutant proteins, using an antibody against the MYC-tag. WT-CARM1, but none of the non-phosphorylatable CARM1 mutants, were able to pull down RUNX1 (Fig. 4B), confirming that CARM1 phosphorylation is required for its ability to interact with RUNX1.

## RUNX1 R223 and R319 are strongly methylated by phosphorylated CARM1

Next, we examined whether CARM1 phosphorylation affected its methylation of RUNX1 using mass spectrometry and an in vitro methylation assay. First, we confirmed R223 as a CARM1 target site but we also identified arginine-319 (R319) in RUNX1 as another potential CARM1 methylation site. Neither of these sites was methylated by PRMT5 (Supplementary Fig. 8A). We then generated an asymmetric dimethylation-specific anti-RUNX1-R319 antibody, and used this antibody and a previously published asymmetric dimethylation-specific anti-RUNX1-R223 antibody[15] (Supplementary Fig. 8B) to show that RUNX1-R223 and -R319 are indeed methylated by CARM1 in vivo. Both

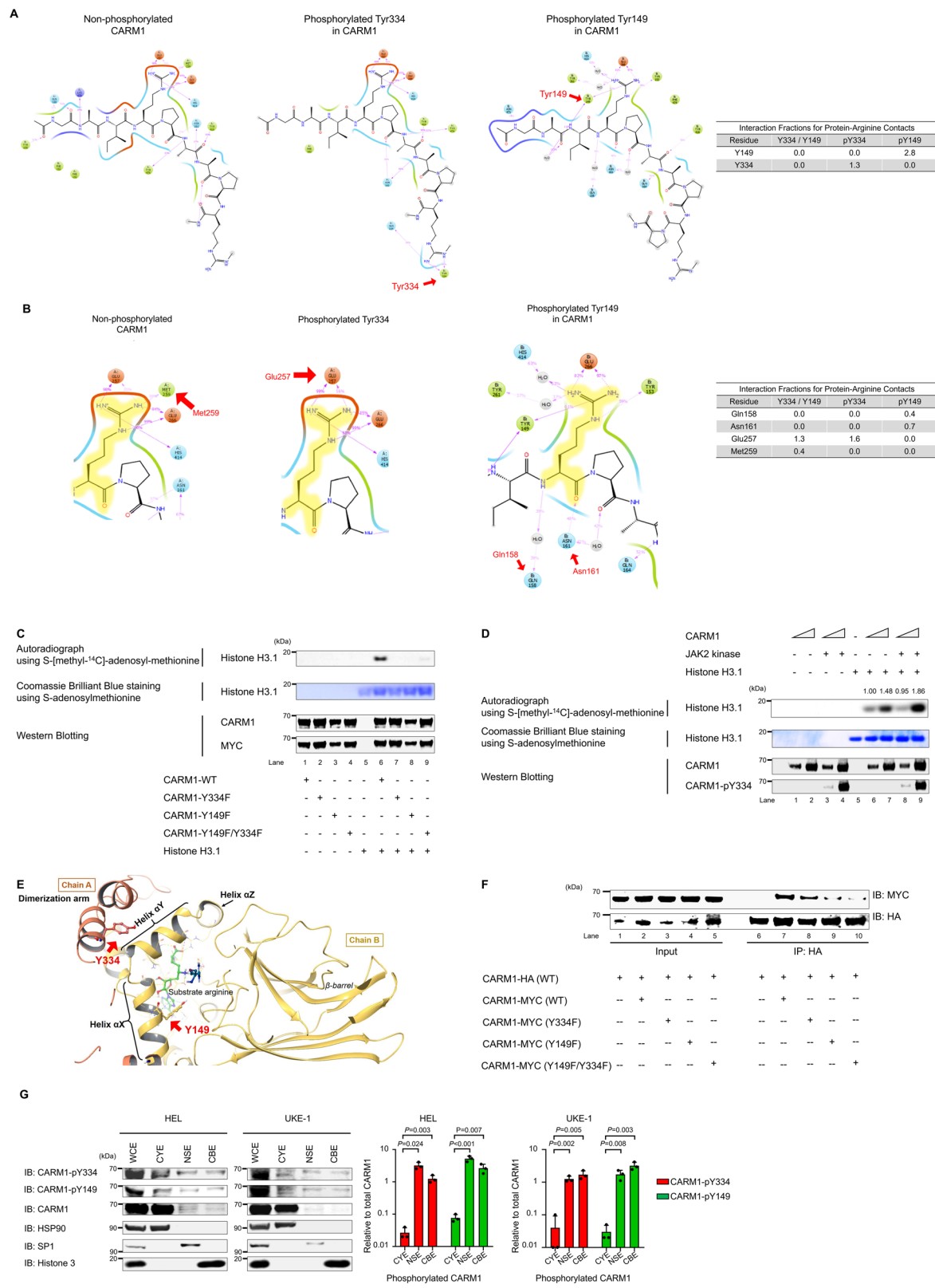

residues, R223 which is located C-terminal to the RUNX1b DNA binding domain and R319 which is located within the activation domain of RUNX1b, are evolutionarily conserved among mammals (Supplementary Fig. 8 C-E).

To prove that CARM1 is the relevant methyltransferase, we then created doxycycline-inducible CARM1 knockdown (KD) HEL cells, using three different small hairpin RNAs (shRNAs) that significantly

decrease CARM1 RNA and protein expression (Supplementary Fig. 9). KD of CARM1 by all three shRNAs significantly decreased the level of RUNX1-R223me2a and RUNX1-R319me2a; they also decreased the ADMA levels of BAF155 and PABP1 (Fig. 4C).

To define the role of CARM1 phosphorylation on the asymmetric dimethylation of RUNX1, and its other substrates, we generated isogenic HEL cell lines carrying homozygous CARM1 non-phosphorylatable

**Fig. 3 | Biochemical regulation of CARM1 enzymatic activity by tyrosine phosphorylation. A** Based on the crystal structures of CARM1, CARM1-Y334 phosphorylation (middle) increased the interaction of the region containing Y334 itself with the substrate compared with non-phosphorylated Y334 (left). CARM1-Y149 phosphorylation (right) also increased the binding of the region containing Y149 itself with the substrate. **B** CARM1-Y334 and -Y149 phosphorylation impairs the loss of binding between CARM1 methionine-259 (Met259) and substrate. The decreased Met259 binding increases the interaction of glutamic acid-257 (Glu257) and glutamic acid-266 (Glu266) with substrates (middle) in the presence of Y334 phosphorylation, compared to non-phosphorylated Y334 (left). Furthermore, the loss of methionine-259 binding increases the interaction of glutamine-158 (Gln158) and aspartic acid-161 (Asn161) with substrates in the presence of Y149 phosphorylation (right). **C** Immunoprecipitation was performed using an anti-MYC antibody and 293 T cells that express MYC-tagged WT or non-phosphorylatable CARM1. The mutant CARM1 Y149F, Y334F, and Y149F/Y334F proteins show reduced methyltransferase activity for histone H3.1, compared to wild-type (WT) CARM1, in in vitro methylation assay. CARM1 and MYC protein lanes are shown to demonstrate equal loading (Top). Autoradiograph of the methylated ³H-histone H3.1 (Middle). Coomassie staining shows histone H3.1 used in the assay (Bottom). Western blotting shows the relative amount of CARM1 and MYC. **D** Recombinant CARM1 protein (produced in *E.coli*), histone H3.1, and ¹⁴C-SAM were incubated to perform an in vitro methylation assay. CARM1 was also incubated with JAK2 kinase, leading to its phosphorylation on Y334 in lanes 3, 4, 7, and 8. Phosphorylated CARM1 shows increased methyltransferase activity for histone H3.1 in in vitro methylation assay (Top). Autoradiograph of the methylated ³H-histone H3.1 (Middle). Coomassie staining shows histone H3.1 used in the assay (Bottom). Western blotting shows the relative amount of CARM1 and phosphorylated CARM1. **E** CARM1-Y149 and -Y334 localize at dimerization arm and helix αX, respectively. These residues lie close to the dimerization interface in the modeled CARM1 structure. **F** Co-immunoprecipitation of HA- and MYC-tagged CARM1 from 293 T cell extracts transiently transfected with plasmid expressing HA-tagged WT and MYC-tagged WT or mutant CARM1. HA-tagged WT CARM1 was immunoprecipitated from cell extracts with anti-HA antibodies, and then the coimmunoprecipitated MYC-tagged CARM1 was probed with anti-MYC antibodies. The levels of MYC-tagged CARM1 Y149F and Y149F/Y334F from the HA immunoprecipitates were lower than those of MYC-tagged CARM1 WT. **G** Subcellular fractionations of HEL cells and UKE-1 cells were immunoblotted using anti-total CARM1, CARM1-pY334, and -pY149 antibodies; cytoplasmic extraction, CYE; nuclear soluble extraction, NSE; and chromatin-bound extraction, CBE. The left lane represents the expression levels of the indicated proteins of whole-cell lysates (WCE). The bar graph on the right represents the ratio of cytoplasmic, nuclear, or chromatin-binding CARM1-pY334 and -pY149 to total cytoplasmic, nuclear, or chromatin-binding CARM1, respectively (bands inside the boxes). Data represent the mean ± SD. *n* = 3, unpaired two-tailed Student's *t*-test. **C, D, F** All experiments were repeated two times independently.

mutation (Y149F or Y334F single mutations, or Y149F/Y334F double mutation), using the CRISPR/Cas9 nuclease system (Supplementary Fig. 10A). We then examined the HEL cells harboring non-phosphorylatable CARM1 mutations (Y149F or Y334F single mutations, or the Y149F/Y334F double mutation), and found that mutation of either tyrosine residue abrogated the asymmetric dimethylation of RUNX1-R223 and RUNX1-R319 (Fig. 4D). HEL cells containing either of the non-phosphorylatable single mutations also showed decreased levels of asymmetrically dimethylated BAF155 and PABP1, while HEL cells with the CARM1 double mutation showed much lower levels of ADMA BAF155 and PABP1 than those with single mutations. In contrast to the enzymatical dead mutant CARM1-E267Q protein, which completely lost the ability to asymmetrically dimethylate BAF155, PABP1, or RUNX1[15,24], the non-phosphorylatable CARM1 mutations maintain enzymatic activity on BAF155 and PABP1, indicating their proper folding. Furthermore, the asymmetrical dimethylation of histone H3R17 and H3R26 was only slightly decreased in HEL cells expressing the CARM1-Y149F/Y334F double mutant protein (Supplementary Fig. 10B), demonstrating the relatively intact enzymatic activity of non-phosphorylatable CARM1 on histone H3.1. Thus, methylation of RUNX1 but not BAF155, PABP1, or histone 3.1 is particularly sensitive to CARM1 phosphorylation status.

To confirm that the JAK2-dependent tyrosine phosphorylation of CARM1 is responsible for the increased ADMA of RUNX1 and other CARM1 substrates in HEL cells, we treated HEL cells with RUX (Fig. 4E). RUNX1-R223 and -R319 dimethylation was reduced after 5 days of RUX exposure. A modest decrease in BAF155 dimethylation and a greater decrease in PABP1 dimethylation were also seen.

To better understand the kinetics of methylation and re-methylation of CARM1 substrates, we treated HEL cells with a selective CARM1 inhibitor (EPZ025654) for 5 days, and monitored substrate methylation over the subsequent five days. While BAF155 and PABP1 methylation was restored after a 1-day EPZ025654-free period, RUNX1 re-methylation was first observed three days after EPZ025654 removal (Supplementary Fig. 11). We next assessed the re-methylation levels of BAF155, PABP1, and RUNX1 in HEL cells treated with RUX (or DMSO) after the removal of EPZ025654. RUX significantly impaired the re-methylation of RUNX1, but not that of BAF155 or PABP1 (Fig. 4F). These results indicate the different requirements and time course of the effect of JAK2 on the ADMA of various CARM1 substrates; RUNX1 dimethylation is sensitive to JAK2 kinase inhibition, while BAF155 dimethylation is only sensitive to CARM1 inhibition.

## Biological consequence of CARM1 phosphorylation

We used the non-phosphorylatable CARM1 mutant knock-in HEL cells to examine the biological effects of CARM-Y149 and -Y334 phosphorylation on cell behavior. While the single mutant cells grew normally, the double CARM1 mutant (Y149F/Y334F)-expressing HEL cells showed reduced proliferation (Fig. 5A). Cell cycle analysis revealed that all three cell lines (harboring homozygous single or double mutant CARM1) had a decreased S-phase fraction and an increased G2/M fraction (consistent with significant G2/M arrest) (Fig. 5B). Having previously shown that CARM1 depletion induced G0/G1 arrest[24], the non-phosphorylatable CARM1 mutants have a distinct effect on cell cycle progression compared to CARM1 depletion. We also observed an increase in apoptosis (indicated by an increased sub-G1 fraction and annexin V-positive cells) particularly in cells harboring the CARM1-Y149F/Y334F double mutation (Fig. 5B and Supplementary Fig. 12). These results demonstrate that phosphorylation of Y149 and Y334 in CARM1 regulates the proliferation of HEL cells, affecting both cell cycle and apoptosis to varying degrees.

## Distinct transcriptional profiles regulated by phosphorylated CARM1

To understand the molecular consequences of CARM1 phosphorylation, we examined the gene expression profiles of the non-phosphorylatable CARM1 mutation knock-in cells, by RNA sequencing. We identified 1,505 differentially expressed genes (≥1.5-fold change and adjusted *p*-value < 0.05) between the CARM1 WT and -Y149F/Y334F double mutant cells (Fig. 5C), while the single CARM1-Y149F or Y334F knock-in cells showed fewer differentially expressed genes, when compared to CARM1 WT cells (Supplementary Fig. 13A, B). Gene ontology (GO) analysis identified that genes involved in G2/M cell cycle progression and apoptosis were negatively enriched in Y149F/Y334F double mutation knock-in HEL cells (Fig. 5D). We also used gene set enrichment analysis (GSEA) to identify pathways differentially regulated by the lack of CARM1 phosphorylation and found that gene sets associated with G2/M cell cycle progression and anti-apoptosis were significantly downregulated in Y149F/Y334F mutation knock-in HEL cells (Fig. 5E). Heatmaps of FDR (*q* < 0.25) values in three non-phosphorylatable CARM1 mutation knock-in cell lines are shown in Fig. 5F, which show that G2/M checkpoint-associated gene sets were downregulated in cells harboring Y149F/Y334F double and Y149 single mutations, while anti-apoptosis-associated gene sets were downregulated in Y149F/Y334F double mutation cells. Among the genes

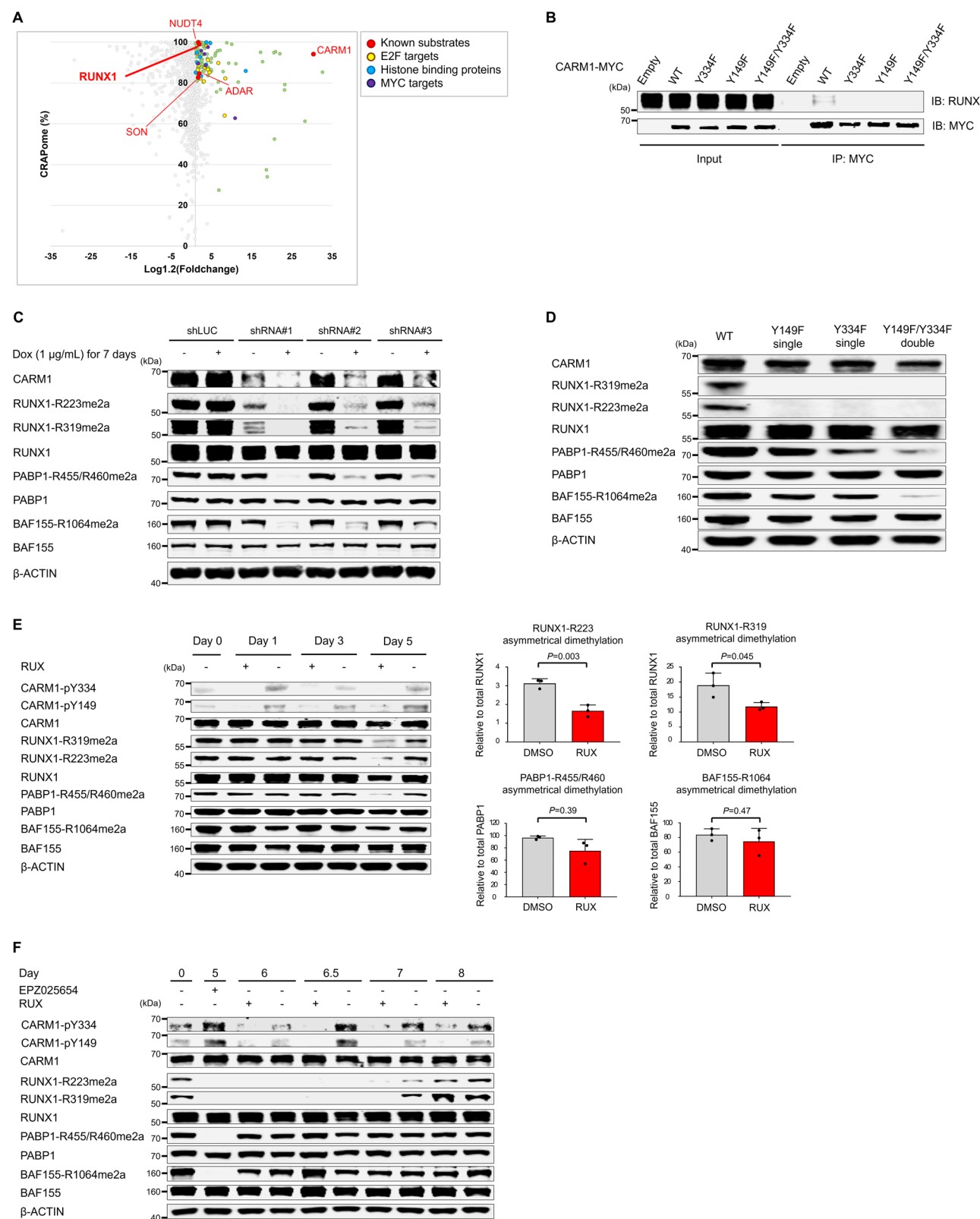

associated with G2/M cell cycle progression and anti-apoptosis that were differentially expressed (adjusted *p* value < 0.05 and FC > 1.5) and relevant in myeloid malignancies[54–59], we found *SMAD3, CCND2, KIF5B, BCL2, BCL2A1,* and *SATB1* (Fig. 5G). We confirmed the downregulation of both G2/M checkpoint (*SMAD3, CCND2,* and *KIF5B*) and anti-apoptosis (*BCL2, BCL2A1,* and *SATB1*) genes in the double CARM1 mutation knock-in HEL cells, using qRT-PCR (Supplementary Fig. 13C, D).

Consistent with our previous reports that CARM1 promotes a DPF2-containing repressor complex that repress miR-223 expression[3,15], we found increased expression of miR-223 in CARM1-Y149F cells and in the CARM1-Y149F/Y334F double mutation cells (Supplementary Fig. 13E). As miR-223 promotes myeloid differentiation, its decreased expression led us to evaluate the expression of HSPC stemness-related genes by RNA sequencing. Gene sets

**Fig. 4 | Identification of RUNX1 as CARM1-interacting proteins by Proximity BioID proteomics. A** Scatter plot comparing mean-fold change for CARM1-BirA* fusion *vs.* BirA* alone with abundance in published negative control AP-MS datasets (%CRAPome). Green dots represent proteins (i) with a cutoff frequency of ≥80% CRAPome and an average spectral count fold change ≥1.2 or (ii) with a cutoff frequency of <80% CRAPome but the average spectral count fold change ≥3.0. Known substrates of CARM1 are indicated as red, and E2F-targets, histone binding proteins, and MYC-targets are shown in yellow, blue, and violet, respectively. See also supplementary data 1 (HEL cells, *n* = 1) and 2 (K562 cells, *n* = 1). **B** Proteins were immunoprecipitated from HEL cell extracts that express MYC-tagged CARM1 (WT and non-phosphorylatable mutants), using an anti-MYC antibody; immunoblotting was then performed using an anti-MYC antibody and anti-RUNX1 mouse antibody. **C** Doxycycline-inducible short hairpin RNAs (shRNAs) directed against CARM1 decreased CARM1 protein levels and the ADMA levels of RUNX1-R223 and -R319 as well as well-established targets, such as PABP1-R455/R460 and BAF155-R1064.

**D** Clustered regularly interspaced short palindromic repeat (CRISPR)/CRISPR-associated protein-9 (Cas9)-mediated non-phosphorylatable CARM1 mutants decreased the ADMA levels of RUNX1-R223 and -R319 as well as PABP1 and BAF155. **E** Expression of total and asymmetry dimethylated RUNX1, PABP1, and BAF155 were assessed in HEL cells treated with RUX 250 nM or DMSO control for 5 days. Fresh media with RUX or DMSO was added on days 0, 2, and 4. Quantification of the ADMA levels of RUNX1, BAF155, and PABP1 at 5 days after RUX treatment are shown in the right panels. Data represent the mean ± SD. *n* = 3, one-way ANOVA. The relative cell viability at day 5 was 85% for cells treated with RUX, compared to those cultured with DMSO (*P* = 0.005)(*n* = 3, biological replicates). **F** The levels of ADMA RUNX1, BAF155, and PABP1 were measured in HEL cells treated with RUX (or DMSO) after EPZ025654 treatment for 5 days followed by a wash-out phase lasting up to 3 days (labeled as day 8). The relative cell viability at day 8 was 86.8% for cells treated with RUX compared to those cultured with DMSO (*P* = 0.006)(*n* = 3 biological replicates). The experiments were repeated at least two times independently. **B**, **D** *n* = 2, (**C**) *n* = 3.

associated with HSPC stemness showed decreased expression in CARM1-Y149F/Y334F mutation knock-in HEL cells (normalized enrichment score of −1.36 with FDR q-value of 0.216) (Fig. 5H), and we confirmed that two HSPC-associated genes, *CD34* and *BMI-1*, were downregulated in Y149F single and Y149F/Y334F double mutation knock-in cells, based on qRT-PCR (Fig. 5I). RUNX (RUNX1, 2, and 3)-target gene sets were not significantly altered in CARM1-Y149F/Y334F mutation knock-in HEL cells (FDR q-value of 0.717) (Supplementary Fig. 13F); however, three RUNX-target genes (*ID2*, *MIR144*, and *RNF144A*) were identified as a subset of core-enrichment genes with ≥ 1.5-fold change and adjusted *p*-value < 0.05. Given that RUNX1 regulates the transcription of *ID2* and *MIR144*[60,61], we independently evaluated their gene expression by qRT-PCR, and confirmed that the Y149F/Y334F double mutation knock-in induced the upregulation of *ID2* and *MIR144* mRNA (Fig. 5J).

To investigate the global localization of RUNX1 on chromatin, we performed ChIP-seq analyses using antibodies against asymmetrically dimethylated R319-RUNX1 and total RUNX1. As we expected, knock-in of CARM1-Y149F or -Y334F single, or -Y149F/Y334F double mutation decreased the overall signal of chromatin-bound dimethylated RUNX1-R319 and, to a lesser degree, total RUNX1 (Fig. 5K). We found that dimethylated R319-RUNX1 shared occupancy for *ID2*, *MIR144*, and *MIR223* with total RUNX1 in HEL cells expressing CARM1-WT (Fig. 5L and Supplementary Fig. 14). In addition, HEL cells carrying CARM1 non-phosphorylatable mutations showed decreased signals for dimethylated RUNX1-R319 within 5 kb of the transcription start sites for *ID2*, *MIR144*, and *MIR223* with less effect on total R319-RUNX1 occupancy. These results demonstrate that the chromatin binding of asymmetrically dimethylated R319-RUNX1 and to a lesser extent unmethylated RUNX1 is regulated by phosphorylated CARM1, allowing the JAK2-CARM1 signaling axis to selectively regulate RUNX1 target-gene expression.

### Targeting the JAK2-CARM1 signaling axis
We confirmed that CARM1 KD or inhibition reduced the proliferation of AML cells (Supplementary Fig. 15A), inducing G0/G1 cell cycle arrest (Supplementary Fig. 15B) and differentiation, and to a lesser degree apoptosis (Supplementary Fig. 15C), consistent with our previous report[24].

We assessed the efficacy of the CARM1 inhibitor, EPZ025654, on a variety of cell lines and found a dose-dependent reduction in the proliferation of UKE-1 cells, but not HEL cells. The half maximal inhibitory concentration (IC50) values of EPZ025654 were 23.7 μM, 1.8 μM, and 186.5 nM in HEL, UKE-1, and SET2 cells, respectively (Supplementary Fig. 16A), while the IC50 values for RUX were 492 nM, 269 nM, and 56 nM in HEL, UKE-1, and SET2 cells, respectively (Supplementary Fig. 16B). We assessed the level of asymmetric dimethyl arginine (ADMA) RUNX1, BAF155, and PABP1 in three cell lines and found that EPZ025654 significantly reduced the levels of ADMA RUNX1 (at R223

and R319)(Supplementary Fig. 16C); it also reduced ADMA BAF155 and PABP1 levels, in a time-dependent manner, without affecting the phosphorylation of JAK2, CARM1, STAT5, ERK, or AKT (Fig. 6A). This suggests that CARM1 inhibitors and kinase inhibitors inhibit cell growth via distinct signaling pathways. To determine whether the combination of EPZ025654 and RUX had synergistic effects on these AML cell lines, we examined cell proliferation after 4 days of EPZ025654 single treatment and subsequently 2 days of combination treatment with RUX and EPZ025654. We observed a significant synergistic inhibition effect on HEL and UKE-1 cells (based on a positive Bliss score), but only an additive effect on SET2 cells (Fig. 6B and Supplementary Fig. 16D), suggesting that the synergistic effect of JAK2 inhibition and CARM1 inhibition (with EPZ025654) may occur primarily in cells that contain phosphorylated CARM1.

We also checked the efficacy of single *vs.* combination therapy on the colony-forming capacity of these cell lines, and found a significant reduction in the colony-forming potential of HEL and UKE-1 cells using combination therapy for 14 days, compared with either inhibitor alone; this effect was not seen in SET2 cells (Fig. 6C). These results provide further evidence that the inhibition of JAK2 has the potential ability to sensitize cells expressing phospho-CARM1 to CARM1 inhibition.

## Discussion
Having identified the phosphorylation of Y149 and Y334 in CARM1 as additional PTMs mediated by the JAK2-V617F mutant kinase, we show that these PTMs increase the enzymatic activity and alter the cellular localization and target specificity of CARM1. CARM1 phosphorylation enhances its ability to block differentiation, and regulate apoptosis and cell cycling by controlling G2/M checkpoints (Fig. 7). Our work highlights the importance of the regulatory effects of JAK2-V617F on the phenotype driven by CARM1 in hematologic cells.

We have demonstrated that the auto-phosphorylation (and cross-phosphorylation) of Y1007/Y1008 in the JAK2 activation loop (by JAK2, JAK1, or TYK2) strongly promotes CARM1 tyrosine phosphorylation, especially in bi-allelic JAK2-V617F mutant cells. Once phosphorylated JAK2 is able to phosphorylate CARM1, however, based on the relatively weak binding of CARM1 and JAK2, it seems likely that JAK2 phosphorylates CARM1 in a hit-and-run manner, rather than via a prolonged interaction. We have found that CARM1-Y149 and -Y334 phosphorylation is promoted by the active conformation of the mutant JAK2 protein, which is stimulated even in the case of JAK2-V617F by the expression of type I cytokine receptors (e.g. EpoR, MPL, or G-CSFR), and inhibited by prolonged exposure to type I JAK2-inhibitors[51,52]. Increased CARM1 phosphorylation appears to be a biological marker of cells with hyperactivated JAK2 (e.g. the JAK2-V617F mutant protein)[62], while KD of either JAK1 or TYK2 decreased the level of phospho-CARM1 by altering the ability of JAK2 to phosphorylate CARM1. In addition, the presence of phospho-Y149 CARM1 in cell lines that lack JAK2V617F (e.g., NB4 cells or SKNO-1 cells, Fig. 2A) suggests that

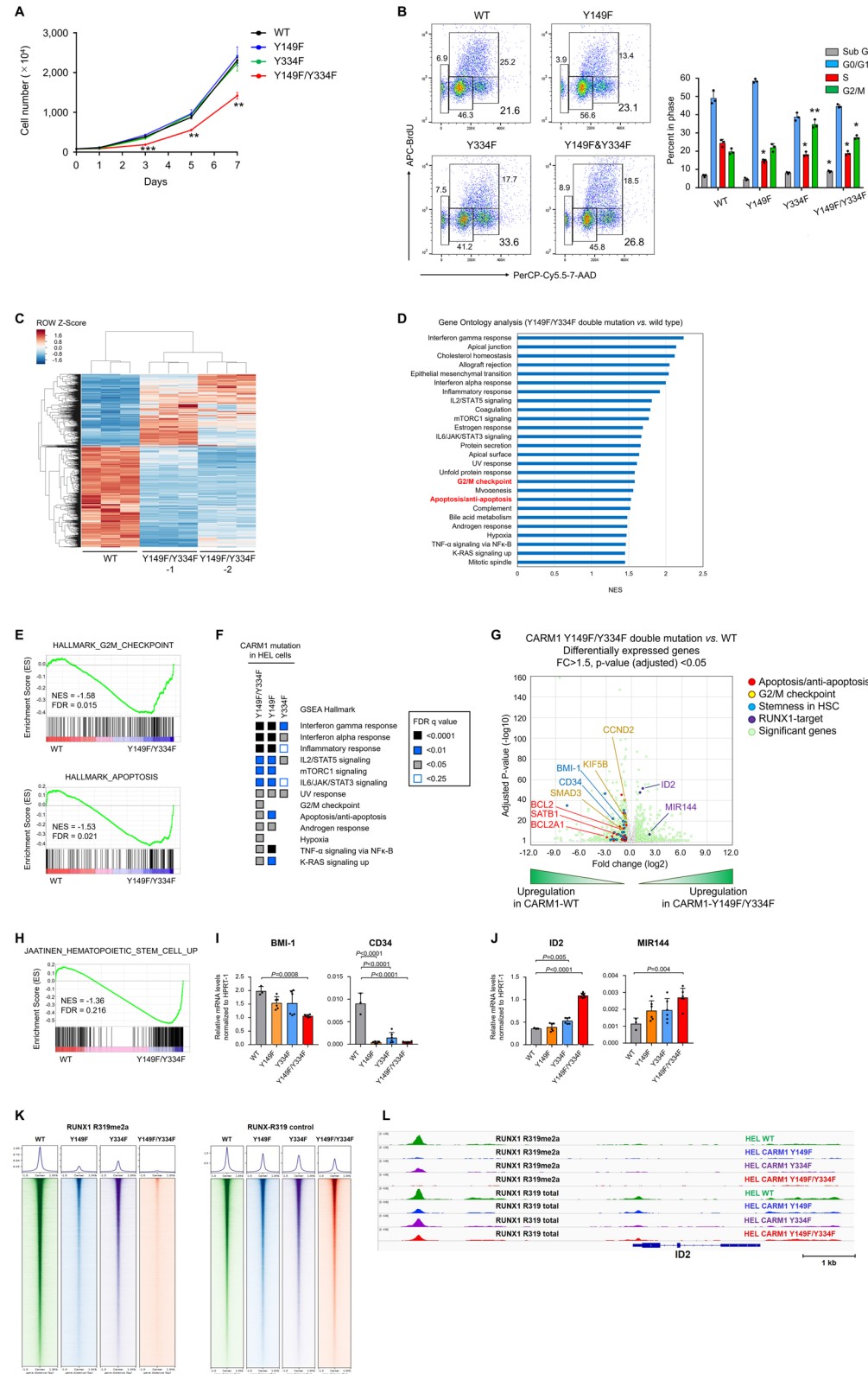

that other kinases are also capable of phosphorylating CARM1 in AML cells.

We observed the differential methyltransferase activity of CARM1 on histone 3.1, RUNX1, and the other substrates when CARM1 was phosphorylated, and the reduced methylation of CARM1 substrates in cells harboring Y149F and/or Y334F CARM1 mutations, which render CARM1 non-phosphorylatable. Our crystal structural model analysis

shows that Y149 and Y334 phosphorylation increases CARM1 binding to its substrates. Phosphorylation of Y149 and Y334 in CARM1 also promotes its nuclear localization, and its methylation of nuclear histone, chromatin binding, and RUNX1 proteins. The Y149F mutant form of CARM1 in particular, showed diminished CARM1 dimerization and minimal methyltransferase activity. We confirmed RUNX1 as a key protein interaction with CARM1, and showed that the non-

**Fig. 5 | Functional analysis of non-phosphorylatable mutant CARM1. A** Cell proliferation assays of non-phosphorylatable CARM1 mutant knock-in HEL cells, where cell numbers were measured using a cell-counting apparatus. Y149F/Y334F ($n = 3$) vs. WT type ($n = 3$); day 3 ($p < 0.001$), day 5 ($p = 0.003$), day 7 ($p = 0.001$). **B** The flow cytometry analysis of BrdU-stained HEL cells expressing non-phosphorylatable CARM1 mutants. Mean fractions ± s.d. in sub G1, G0/G1, S, and G2/M populations. $n = 3$. S population ($p = 0.015$) in Y149F vs. WT; S ($p = 0.011$) and G2/M populations ($p = 0.009$) in Y334F vs. WT; S ($p = 0.008$) and G2/M populations ($p = 0.002$) in Y149F/Y334F vs. WT. **C** Heatmap shows the differentially expressed coding genes at 2-fold cut-off, representing replicates of HEL cells expressing CARM1 WT or two independent cells expressing CAMR1-Y149F/Y334F double mutation (Y149F/Y334F-1 and Y149F/Y334F-2). **D** Gene ontology analysis of significant downregulated genes in HEL cells expressing CARM1-Y149F/Y334F compared to CARM1-WT. **E** Heatmaps of FDR ($q < 0.25$) values from GSEA of hallmark gene set collections. **F** Representative GSEA plot depicting the downregulation of G2/M checkpoint and apoptosis/anti-apoptosis pathways. **G** Volcano plot representing gene expression changes triggered by CARM1-Y149F/Y334F mutation knock-in in HEL cells. Genes associated with apoptosis/anti-apoptosis, G2/M checkpoints, stemness in hematopoietic stem cells, and RUNX1-target are shown in red, yellow, blue, and violet, respectively. The red dots indicate upregulated genes in HEL cells expressing CARM1-Y149F/Y334F, whereas the blue dots indicate downregulated genes. *P* values correspond to a two-sided Wilcoxon rank-sum test with Bonferroni correction. **H** Representative GSEA plot depicting the downregulation of "hematopoietic stem cell up" signature. **I** qRT-PCR analysis showing *BMI-1* and *CD34* in HEL cells expressing CARM1 WT ($n = 3$), Y149F ($n = 6$), Y334F ($n = 6$), and Y149F/Y334F mutation ($n = 6$). Mean and SD are expressed as a percentage of *HPRT-1* expression. **J** qRT-PCR analysis showing *ID2* and *MIR144* in HEL cells expressing CARM1 WT ($n = 3$), Y149F ($n = 6$), Y334F ($n = 6$), and Y149F/Y334F mutation ($n = 6$). Mean and SD are expressed as a percentage of *HPRT-1* expression. $n = 3$. **K** Heat map of total R319-RUNX1 or asymmetrically dimethylated R319-RUNX1 binding tag intensity by ChIP-seq analysis for HEL cells expressing CARM1 WT, Y149F, Y334F, or Y149F/Y334F mutant proteins. **L** ChIP-seq analyses were performed to assess total RUNX1 and asymmetrically dimethylated R319-RUNX1 chromatin binding. Target occupancies at the *ID2* gene are shown in IGV genome browser tracks. All error bars represent the mean ± SD. *P* values were determined by two-tailed Student's *t*-test (**A**, **B**) and one-way ANOVA followed by Dunnett's post hoc test (**I**, **J**). *$p < 0.05$, **$p < 0.01$, ***$p < 0.001$.

phosphorylatable CARM1 mutant proteins did not bind RUNX1. Thus, in addition to promoting dimerization, phosphorylation of CARM1 affects its localization, substrate binding, and methyltransferase activity.

Having previously shown that the enzymatic activity of CARM1 is required to dimethylate RUNX1-R223[15], we have now identified R319 of RUNX1 as a second arginine residue asymmetrically dimethylated by CARM1 (but not PRMT5) and showed that CARM1-Y149 and -Y334 phosphorylation enhanced the asymmetrical dimethylation of both R223- and R319-RUNX1. CARM1 can regulate hematopoietic cell differentiation through multiple mechanisms, including the generation of a repressor complex that contains asymmetrically dimethylated RUNX1-R223 and negatively regulates miR-223 expression[3,15,63]. Consistent with that effect, we found that non-phosphorylatable CARM1 mutant cell lines show increased expression of miR-223 and several other RUNX1-target genes (*ID2* and *MIR144*). Knock-in of a non-phosphorylatable CARM1 mutation also downregulated the expression of BMI-1, which is a regulator of self-renewal that plays a role in JAK2-V617F mutant hematopoietic stem cells[64], as well as other cancer stem cell phenotypes. Given its substrate targets (e.g. BAF155 and RUNX1) and gene targets (e.g. *BMI-1* and *ID2*), CARM1 and phospho-CARM1 appear to play a pivotal role in hematologic malignancies with the *JAK2*-V617F mutation, and likely in other settings as well.

The non-phosphorylatable CARM1 mutation (Y149F/Y334F) knock-in HEL cells show decreased cell growth with increased cell cycle arrest and apoptosis, likely due to the downregulation of gene expression associated with G2/M progression and anti-apoptosis, including BCL2 family members (*BCL2* and *BCL2A1*) which have been implicated in JAK2-V617F expressing myeloid malignancies[65-67]. Similarly, the asymmetrical dimethylation of BAF155 by CARM1 has been shown to inhibit the apoptosis of ovarian cancer cells through downregulation of pro-apoptotic gene expression (*DAB2*, *DLC1*, and *NOXA*)[68]. We could not detect a significant difference in the expression of these genes in WT- *vs.* Y149F/Y334F-CARM1 expressing HEL cells, despite similar changes in the level of ADMA BAF155, confirming the cell-context-specific effects of arginine dimethylation, which may relate to homeostatic mechanisms that control cell survival.

The dependency of JAK2-V617F mutant AML cells on CARM1 is consistent with our previous studies showing that CARM1 is an essential gene for the growth of myeloid leukemia cells. Furthermore, a genome-wide CRISPR/Cas9 knockout screen conducted as part of the Dependency Map database (https://depmap.org/portal/) also identified the CARM1-dependency of JAK2-V617F mutant cell lines[69]. Non-phosphorylatable CARM1 mutant-expressing HEL cells showed significantly decreased cell growth, suggesting some dependency of HEL cells on CARM1 phosphorylation. Indeed, when we evaluated the efficacy of inhibiting both JAK2 and CARM1, we found that small-molecule inhibitors targeting CARM1 (EPZ025654) sensitized JAK2-mutant cells to JAK2 inhibition, and that targeting JAK2 increased the sensitivity of JAK2-V617F-positive cells to CARM1 inhibition.

Our studies demonstrated how JAK2 signaling differentially affects the functions of epigenetic writers, such as CARM1 and PRMT5, and the PTM of histones as well. Phosphorylation of PRMT5 mediated by JAK2-V617F impairs its binding to MEP50 which abrogates its symmetric arginine methyltransferase activity on histones[48]. The present study showed that auto-phosphorylation and cross-phosphorylation of JAK2-V617F mutants triggers CARM1 phosphorylation, increasing its nuclear localization and promoting cell survival and cell cycle progression. Although the phosphorylation of CARM1 does not seem necessary for CARM1 to methylate histones, it is essential for RUNX1 arginine methylation. Phosphorylation of histone H3.1 (Y41) and TET2 (Y1939/Y1964) mediated by JAK2 also affects gene activation *vs.* repression at numerous gene regulatory regions[46,47]. This suggests that the JAK2-V617F kinase alters each epigenetic writer differently, allowing for great diversity in its effects on downstream substrate proteins. Given that high levels of JAK2-V617F mutant phosphorylation correlates with shorter survival[70] and resistance to JAK1/JAK2 inhibitors[53], combined JAK2 and CARM1 inhibition may have therapeutic potential for certain patients, perhaps those with high phospho-CARM1 levels, although specific CARM1 inhibitors are not in clinical trials at this time.

In conclusion, CARM1 phosphorylation mediated by hyper-activated JAK2 regulates its methyltransferase activity, localization, and substrate specificity. In certain settings, for example in cells with high levels of phospho-JAK2, CARM1 phosphorylation appears to be required for maximal proliferation of myeloid neoplasms. Our results suggest a potential role of targeting both JAK2 and CARM1 in JAK2-V617F mutant myeloid malignancies.

## Methods

### Cell lines and cell culture

Leukemia cell lines were purchased from the ATCC, DSMZ, or Coriell, and cultured according to the provider's instructions (Supplementary Table 1). All cell lines were grown at 37 °C in 5% $CO_2$: HEL cells were grown in RPMI 1640 media with 10% fetal bovine serum (FBS) (Invitrogen). SKNO-1 cells were supplemented with 10 ng/mL GM-CSF (Peprotech); TF-1 cells were supplemented with 10 ng/mL IL-3 (Peprotech); and UKE-1 cells were supplemented with 10% fetal calf serum, 10% horse serum, and 1 µM hydrocortisone (Sigma). 293 T cells were grown in 10 cm dishes with 10% FBS and DMEM media. HEL, SET2, and UKE-1 cells were treated with a selective JAK1/JAK2 inhibitor

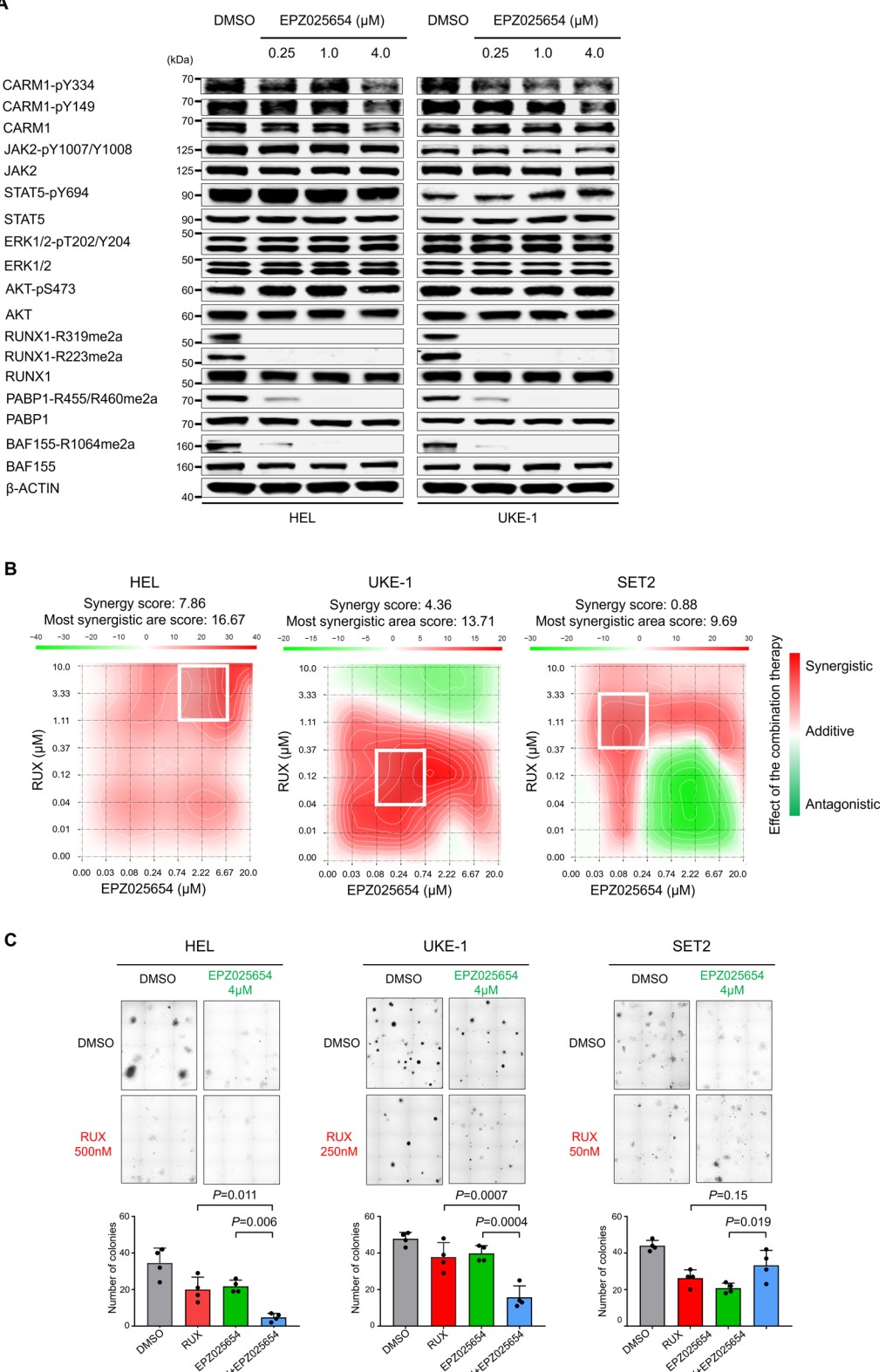

**Fig. 6 | Inhibition of CARM1 targets cells harboring phosphorylated CARM1 mediated by JAK2-V617F mutant. A** Western blot assessment of phosphorylation in JAK2, STAT5, ERK, and AKT, and asymmetric demethylated arginine in RUNX1, BAF155, and PABP1 in HEL and UKE-1 cells treated with 5 days with increasing concentrations of EPZ025654 (μM). These experiments were repeated independently twice. **B** Excess over Bliss plots (Bliss method) showing synergistic effects between EPZ025654 and RUX were visualized in the calculated 2D synergy maps. Red and green areas represent synergistic (synergy score >+10), additive (synergy score 0- +10), and antagonistic effect (<−10), respectively. In 2D synergy maps, white rectangles show the maximum synergy area in each cell. **C** The colony formation of HEL, UKE-1, and SET2 cells treated with DMSO (control), RUX, EPZ025654, or a combination of RUX and EPZ025654. The concentration of RUX was applied based on the IC50 values for each cell line. Representative pictures of colonies on semi-solid methylcellulose media are shown on the upper panels. Quantification of the number of colonies at 14 days after plating are shown in the lower panels. Data represent the mean ± SD. $n = 4$, one-way ANOVA.

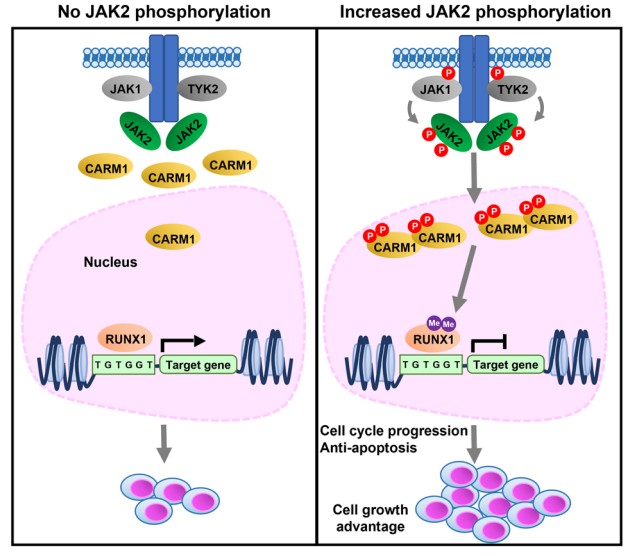

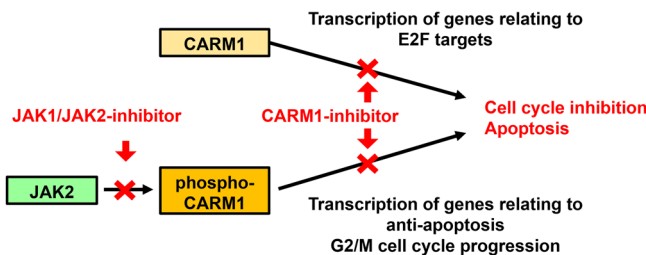

**Fig. 7 | A schematic model showing JAK2-CARM1 axis.** JAK2-V617F mutant kinase, when activated by JAK2, JAK1, or TYK2, strongly phosphorylates CARM1-Y149 and -Y334, increasing its methyltransferase activity and the asymmetrical dimethylation of its substrates, including histone 3 and RUNX1. CARM1 phosphorylation promotes cell-cycle progression and inhibits apoptosis, and regulates the genes associated with stemness (BMI-1).

(ruxolitinib, RUX) at the indicated concentration, in order to inhibit JAK2 activity (DMSO was used as the control).

## Patient samples

Bone marrow samples were collected from patients with myeloid neoplasms diagnosed at Sylvester Comprehensive Cancer Center. All samples were obtained during the routine clinical care of the patients and were de-identified prior to analysis. The unique patient numbers (UPNs) were assigned for this study. Next-generation sequencing was performed on DNA extracted from bone marrow samples, using the next-generation sequencing panel (FoundationOne Heme, GenPath OnkoSight Myeloid, Genoptix NexCourse Complete, Genoptix AML Molecular Profile, Genoptix Myeloid Molecular Profile, and Illumina TrySight Myeloid-54), with genomic alterations (substitutions, small insertions, and deletions), gene-level focal copy number alterations, and structural rearrangements, identified as previously described[71]. The clinical information of patients is shown in Supplementary Table 2.

## Plasmid construction

The full-length sequence of CARM1 was verified by Sanger sequencing[24] and reconstructed into the pCDH-MSCV-EF1 vector, purchased from SBI Biotech. Non-phosphorylatable CARM1 tyrosine to phenylalanine mutant cDNAs (Y149F, Y334F, and Y149F/Y334F) were generated using PCR-based site-directed mutagenesis and confirmed by sequencing analysis. To generate vectors for the BioID system, pCDH-MSCV-EF1 lentiviral vector containing a green fluorescent protein (GFP) reporter gene, a puromycin selection cassette, a fusion with

Multiple Approaches Combined (MAC)-tagged versions (N-terminal: BirA* (Arg118Gly)/HA/StrepIII) and the bait protein (CARM1) were synthesized by In-Fusion® HD Cloning Plus Kit (Takara, #638910).

The pCDH plasmids, expressing one of three human CARM1 shRNAs, were purchased from Sigma: sequences of shRNAs are shown in Supplementary Table 3[72]. Stable knockdown cells were selected with 1 µg/mL puromycin.

## Lentivirus and retrovirus production, concentration, and infection

Lentiviruses were produced in 293 T cells using lipofectamine 2000, psPAX2, and pMD2.G as transfection reagent, packaging plasmid, and envelope plasmid, respectively, according to standard protocols. Lentiviruses were collected 48 hours after transfection and concentrated using the lenti-X concentrator (Clontech). Transient transfection of cell lines was conducted using polybrene (Millipore, TR-1003-G), according to the manufacturer's instructions. To generate stable cell lines, HEL cells were transduced with scrambled shRNA and CARM1 shRNAs expressing vectors, using polybrene. Stable knockdown cells were selected with 1 µg/mL puromycin. Cells were cultured with or without doxycycline induction for 7 days, to induce expression of the shRNA.

## CRISPR/Cas9 base editing in HEL cells

HEL cells expressing clustered regularly interspaced short palindromic repeat (CRISPR)/CRISPR-associated protein-9 (Cas9) mediated knockout (KO) (#4143204-4 and −5) of JAK1 or TYK2, and non-phosphorylatable Y149F, Y334F, and Y149F/Y334F CARM1 mutants-knock-in (#4143204-1, −2, and −3) were generated by Synthego Corporation (Redwood City, CA, USA). To generate these cells, ribonucleoproteins containing the Cas9 protein and synthetic chemically modified sgRNA were electroporated into the cells using Synthego's optimized protocol. Editing efficacy is assessed upon recovery, 48 hours post electroporation. Genomic DNA is extracted from a portion of the cells, PCR amplified, and sequenced Sanger sequencing. The resulting chtomatograms are processed using Synthego Inference of CRISPR edits software (ice.synthego.com). All sgRNA sequences are given in Supplementary Table 4.

To create monoclonal cell populations, edited cell pools are seeded at <1 cell/well via limiting dilution into 96 well plates. All wells are imaged every 3 days to ensure single cell clone expansion. Clonal populations are screened and identified using the PCR-Sanger-ICE genotyping strategy described above.

## In vitro kinase assay

In vitro kinase assays were performed using commercially available JAK2 kinase (Abcam, ab42619), which contains only the kinase domain (from amino acids 808 to 1132), and bacterially-purified GST-CARM1 (Reaction Biology, HMT-11-120) (Supplementary Table 4). Bacterially purified GST-PAK1 (Abcam, ab177574) was used as the control substrate. For each reaction, 0.2 or 1.0 µg of JAK2, 2.0 µg of GST-CARM1 or GST-PAK1, and ATP (Cell Signaling Technology, #9804) were added to the kinase buffer (Cell Signaling Technology, #9802). Reactions were incubated at 30 °C for 20 min, and the proteins were resolved by gel electrophoresis.

## In vitro methylation assay

Wild-type (WT) and non-phosphorylatable GST-CARM1 were incubated with JAK2 kinase, as described in the in vitro kinase assay, and used for in vitro assays. We also used MYC-tagged WT CARM1 and non-phosphorylatable mutant CARM1, which were purified from transfected 293 T cells by an anti-MYC immunoprecipitation. Two known CARM1 substrates, histone H3.1 (New England Biolabs, M2503S) and RUNX1b (MyBioSource, MBS1471613), were used in the methylation assay (Supplementary Table 5). The MYC-CARM1 protein or

recombinant CARM1 protein was then incubated with substrates in the presence of 5 mM Tris [pH 8.5], 2 mM KCl, 1 mM MgCl2, 0.1 mM β-mercaptoethanol, 10 mM sucrose) and S-adenosyl-L-[methyl-3H] methionine (SAM)(15 Ci/mmol, 66 µM; Amersham Bioscience) at 30 °C for 4 h. The reactions were stopped by adding an SDS loading buffer, and the proteins were separated in 4-12% NuPAGE Bis-Tris Protein Gels (Invitrogen, NP0336BOX). After fixation in 45% methanol and 10% acetic acid for 30 min, the gels were treated with Gel Dry Drying Solution (Thermo Fisher Scientific, LC4025) for 15 min, and exposed to x-ray films.

## Subcellular fractionation assay

Subcellular fractionation of HEL, UKE-1, SET2, and K562 cells was performed using the Subcellular Protein Fractionation Kit for cultured cells (Thermo Scientific, #78840) according to the manufacturer's instructions. Briefly, cellular proteins from an equal number of cultured cells were fractionated into cytoplasmic, nuclear soluble, and chromatin-bound fractions. The purity of the subcellular fractions was assessed by blotting with HSP90 (cytoplasmic extraction, CYE), SP1 (nuclear soluble extraction, NSE), and histone H3 (chromatin-bound extraction, CBE).

## Immunoprecipitation studies

Cultured cells were washed (or scraped off plates for 293 T cells) in cold PBS. Cells pellets were resuspended in lysis buffer containing 50 mM Tris-HCl, 150 mM NaCl, 1% Triton X-100, 1 mM EDTA, 1 mM DTT, 1 mM PMSF, and both protease inhibitors and phosphatase inhibitors (Roche). Proteins were precipitated using anti-phosphotyrosine antibody (Millipore Sigma, 05-321X) plus protein A/G agarose beads (Thermo Fisher Scientific, #88802), anti-HA magnetic beads (Thermo Fisher Scientific, #88837), or anti-MYC beads (Thermo Fisher Scientific, #88842) with sonication. Insoluble debris was removed by centrifugation, and then an equal amount of cell lysate was subjected to incubation with magnetic beads at 30 °C for overnight. Following four washes and protein separation using SDS-PAGE gels, the membranes were probed using specific antibodies, as described in the western blotting paragraph.

## Western blotting

Protein samples were separated by electrophoresis on denaturing 4-12% NuPAGE Bis-Tris Protein Gels and blotted to nitrocellulose membranes (Bio-Rad, 1620112). Membranes were blocked in TBS blocking buffer (LI-COR Biosciences, 927-50003). Antibodies were added to directly to the TBS blocking buffer (anti-CARM1-pY149 antibody [1:500], anti-CARM1-pY334 antibody [1:500], anti-RUNX1-R223me2a antibody [1:500], and anti-RUNX1-R319me2a antibody [1:500]) and incubated overnight at 4 °C with gentle shaking. The other antibodies used for immunoblotting and their dilutions are shown in Supplementary Table 6. The unprocessed scans of the important blots are shown in Supplementary Fig. 18-23.

## Flow cytometry

The BrdU assay was performed using the BD Pharmingen BrdU Flow Kit (BD Pharmingen, #552598). Cell surface Annexin V and intracellular 7-AAD staining were performed using an PE Annexin V Apoptosis Detection Kit I (BD Pharmingen, #559763). These assays were performed according to the manufacturer's instructions. Flow cytometry was performed using a FACSCAN running CellQuest software (BD), with data analysis performed using FlowJo (Tree Star, Inc.).

## RNA isolation and quantitative RT-PCR

RNA was extracted using Direct-zol RNA Microprep (ZYMO RESEARCH, R2061) and cDNA synthesized by qScript cDNA Super-Mix (Quanta bio, #95048-100), according to the manufacturer's

instructions. Quantitative RT-PCR was performed using the TaqMan Universal PCR Master Mix (Thermo Fisher Scientific, #4364340). The thermal cycle conditions to amplify cDNA were 48 °C for 15 min; and 95 °C for 10 min, followed by 45 cycles of 95 °C for 15 s; 60 °C for 1 min. HPRT-1 was used as an internal control. The catalog numbers of TaqMan primers are shown in Supplementary Table 7. Relative quantification of the genes was calculated using the method ($2^{-Ct}$) as described by the manufacturer.

## RNA-seq assays

RNA-seq library prep was carried out using the Illumina TruSeq Total Total Stranded kit (RS-122-2201) with Ribo-Zero rRNA reduction following the manufacturer's protocol without modification. Paired-end sequencing (75 base pairs) was performed on a NovaSeq 6000. FASTQ data were processed with cutadapt v1.15 (--nextseq-trim=20 -m 18) to remove low quality reads. Gene level counts were estimated using RSEM v1.3.0 and STAR v2.6.0c alignment to the human hg19 transcriptome (GENCODE V19 annotation). RUVseq v1.12.0 was sed adjusted gene counts by removing unwanted variance using exogenous ERCC spike-in RNA. DESeq2 version 1.18.1 and R (version 3.4.1) were used for sample normalization and differential expression analysis with a q value < 0.25 and an FC > 2.0 (Wald test). Heatmaps were generated using variance-stabilized gene counts from DESeq2. For GSEAs, the Wald statistic of each time point compared to hormone-deprived conditions was used as input for the pre-ranked list of GSEA v3.0 on gene sets (-scoring_scheme weighted -nrom meandiv).

## ChIP-seq assays

CARM1 wild-type or non-phosphorylatable mutant-knock-in HEL cells from two independent experiments were collected and subjected to ChIP-sequencing using antibodies against RUNX1 or asymmetrically dimethylated R319-RUNX1. ChIP and ChIP-seq were performed as previously described[73]. In brief, cells were fixed and lysed with sonication buffer (16.7 mM Tris pH8, 167 mM NaCl, 0.1% SDS, 1% Triton X-100, 1 mM EDTA). After sonication with Bioruptor Pico (Diagenode), insoluble debris was removed by centrifuge. The remaining supernatant was incubated with antibody overnight at 4 °C, and then the antibody-chromatin complex was pulled down with 20 µl of magnetic Protein A beads (NEB S1425s). The beads were washed twice with high salt buffer (20 mM Tris pH8, 500 mM NaCl, 0.1% SDS, 1% Triton X-100 and 2 mM EDTA) and Li buffer (10 mM Tris pH8, 0.25 M LiCl, 1% NP40, 1% sodium deoxycholate and 1 mM EDTA), and then suspended in 30 µl buffer (100 mM Tris pH8, 250 mM NaCl, 1 mM EDTA, and 0.1% SDS) with 1 µl of Proteinase K (Invitrogen, AM2546), and incubated at 65 °C for 4 h. The DNA in the supernatant was purified with Agencourt AMPure beads (Beckman Coulter A63880), and used for ThruPLEX DNA-Seq Kit (Takara, #112219) as DNA library construction and sequenced on Novo-seq platform.

The sequencing reads were aligned to the human genome (hg19) using bowtie2 (2.2.6)[74] with the default parameter. The peaks were called with MACS2 (2.1.1.20160309)[75]. Heatmaps and average binding profiles were generated using DeepTools (3.1.3)[76] and NGS plot (2.61)[77].

## Identification of phosphorylated or methylated residues by mass spectrometry analysis

The in vitro kinase and in vitro methylation assays were performed using recombinant proteins, as described above. A band corresponding to phosphorylated CARM1, methylated H3.1, and methylated RUNX1 were excised from the gel and subjected to further separation on a 4-12% NuPAGE Bis-Tris Protein Gels followed by staining with Coomassie Blue Protein Stain. The CARM1, H3.1, and RUNX1 bands were identified by size and were excised from the gels. Excised gels were sent to the Proteomics and Metabolomics Core Facility at Moffitt Cancer Center for further sample preparation and analysis. Briefly, in-gel digestion with trypsin, chymotrypsin, and GluC enzymes was

performed after protein reduction and alkylation to maximize sequence coverage of target proteins. The digested peptides were purified using ZiptipC18 (Millipore), followed by LC-MS/MS analysis. Peptide identifications were performed using the Mascot and Sequest algorithm for searching against the Swiss-Prot human database and summarized in Scaffold software. Dynamic modifications included carbamidomethylation (Cys), oxidation (Met), methylation (Lys/Arg), and phosphorylation (Ser/Thr/Tyr). Experiments were repeated independently at twice.

## BioID system set-up and pull-down experiments

To generate stable cell lines, that inducibly express MAC (multiple approaches combined)-tagged versions of CARM1, HEL, and K562 cells were transfected with the pCDH-MSCV-EF1-CARM1-MAC tag (CARM1-BirA*) or pCDH-MSCV-EF1- MAC tag (BirA* alone) lentiviral vector using polybrene. MAC-tag contains HA-tag, StrepIII-tag, and a bacterial biotin ligase (Arg188Gly mutant, BirA*). Five days after transfection, GFP-positive cells were selected by flow cytometric cell sorting, then the GFP-positive clones were pooled and amplified in 1 µg/mL puromycin for 2 weeks. Stable cells expressing MAC-tag fused to GFP were used as negative controls and processed in parallel to the MAC-tagged CARM1 protein. Cells were cultured in biotin-free media for 24 or 48 h to allow for cell synchronization. The in vivo biotinylation was activated by supplement with 50 µM biotin as a final concentration, for 16 h. Samples were snap-frozen and stored at −80 °C.

The affinity purification of samples was performed as described previously. In brief, cell pellets were thawed in ice-cold lysis buffer #1 (0.5% IGEPAL, 50 mM Hepes, pH 8.0, 150 mM NaCl, 50 mM NaF, 1.5 mM NaVO₃, 5 mM EDTA, 0.1% SDS, supplemented with 0.5 mM PMSF, phosphatase inhibitors, and protease inhibitors). For BioID pull-downs, cleared lysate was obtained by centrifuge and incubated with MagStrep "type3" XT beads 5% suspension (IBA, 2-4090-002) with overnight rotation at 4 °C to pull down all biotinylated proteins. The beads were washed twice with lysis buffer #2 (0.5% IGEPAL, 50 mM Hepes, pH 8.0, 150 mM NaCl, 50 mM NaF, 1.5 mM NaVO₃, 5 mM EDTA, supplemented with 0.5 mM PMSF, phosphatase inhibitors, and protease inhibitors) and 4 times with wash buffer (50 mM Tris-HCl, pH 8.0, 150 mM NaCl, 50 mM NaF, 5 mM EDTA). The beads were then resuspended in 2 × 300 µL elution buffer (50 mM Tris-HCl, pH 8.0, 150 mM NaCl, 50 mM NaF, 5 mM EDTA, 0.5 mM Biotin) for 5 min and the eluates collected into Eppendorf tubes, followed by reduction of the cysteine bonds using 5 mM Tris(2-carboxyethyl)phosphine (TECP) for 30 min at 37 °C and alkylation with 10 mM iodoacetamide. The proteins were then digested to peptides with sequencing grade modified trypsin (Promega, V5113) at 37 °C overnight. After quenching with 10% TFA, the samples were desalted by C18 reversed-phase spin columns according to the manufacturer's instructions. The eluted peptide sample was dried in a vacuum centrifuge and reconstituted to a final volume of 2 µL in 0.1% Formic Acid and 2% CH₃CN. Liquid Chromatography coupled Parallel reaction monitoring (LC-PRM) method was applied to target proteins, including CARM1 and RUNX1.

## Liquid chromatography-mass spectrometry and data analysis

A nanoflow ultra-high performance liquid chromatograph (RSLC, Dionex, Sunnyvale, CA) coupled to an electrospray benchtop orbitrap mass spectrometer (Q-Exactive, Thermo, San Jose, CA) was used for tandem mass spectrometry peptide sequencing experiments. The sample was first loaded onto a pre-column (2 cm×100 µm ID packed with C18 reversed-phase resin, 5 µm, 100 Å) and washed for 8 min with aqueous 2% acetonitrile and 0,04% trifluoroacetic acid. The trapped peptides were eluted onto the analytical column, (C18, 75 µm ID x 25 cm, 2 µm, 100 Å, Dionex, Sunnyvale, CA). The gradient for 120 min was programmed as: 95% solvent A (2% acetonitrile + 0.1% formic acid)

for 8 min, solvent B (90% acetonitrile ~ 0.1% formic acid) from 5% to 15% in 5 min; 15% to 40% in 85 min; then solvent B from 50% to 90% in 2 min; and re-equilibrate for 10 min. The flow rate on the analytic column was 300 nL/min. The spray voltage was 1900 v. The capillary temperature was 275 °C. S lens RF level was set at 40. The top 20 tandem mass spectra were collected in a data-dependent manner following each survey scan. The resolution for MS and MS/MS scans were set at 60,000 and 45,000 respectively. Dynamic exclusion was 15 seconds for previously sampled peptide peaks.

The proteins were identified using both SEQUEST[78] and MASCOT search engines[79]. The relative quantification of peptides was calculated using MaxQuant (version 1.2.2.5). Peaks were searched against the Human entries in the UniProt database (20,151 sequences; released August 2015)[80]. The raw files were processed with the following parameters: including >7 amino acids per peptide, as many as three missed cleavages, and a false discovery rate (FDR) of 0.01 selected for peptides and proteins. Methionine oxidation and peptide N-terminal acetylation were selected as variable modifications in protein quantification. Carbamidomethylation of cysteine and methionine oxidation were selected as variable modifications. Following protein identification and relative quantification with MaxQuant, the data were normalized using iterative rank order normalization (IRON)[81]. Contaminant Repository for Affinity Purification (CRAPome, https://www.crapome.org/)[82] were used as statistical tools to determine a comprehensive characterization of background contamination and identify specific high-confidence interactions from the data of our mass spectrometry. A cutoff frequency of ≥80% was applied in CRAPome. Proteins with ≥1.2 of the average spectral count fold change in cells expressing CARM1-BirA* compared to those expressing BirA* were considered to have an interaction with CARM1. Additionally, proteins with a CRAPome frequency of <80% and with ≥3.0 of the average spectral count fold change in cells expressing CARM1-BirA* compared to those expressing BirA* were included in the further analysis.

Furthermore, we used Metascape (https://metascape.org/gp/index.html#/main/step1)[83] to perform functional enrichment and interactome analyses for Annotation and Visualization (accessed on March 18. 2020). Express analysis was chosen for enrichment and clustering analysis. Metascape Express analysis consists of an automated analysis workflow beginning with identifier conversion and followed by gene annotation (GO/KEGG terms, canonical pathways, and hallmark gene sets), membership search, and enrichment analysis. The network is visualized with Cytoscape ver 3.1.2 with "force-directed" layout and with edge bundled for clarity. The MCODE algorithm automatically extracts densely connected protein complexes from the list of candidates interacting with CARM1.

## Molecular dynamics simulation

The CARM1 co-crystal structure of sinefungin and the PABP1-R455 peptides were downloaded from the Protein Data Bank (PDB: 5dx1) and processed using "Protein Preparation Wizard" in Maestro v11.4. In the crystal structures of CARM1, peptides sequences included known or hypothesized substrates, such as PABP1-R455, H3-R17, H3-R26, EP300-R2142, EP300-R580, EP300-R604, and BAF155-R1064. Bond orders were assigned using the CCD database. Sinefungin was manually edited to S-adenylmethionine using "3-D Builder" in Maestro Suite at minimized state. H-bond assignments were refined by sampled water orientations in Epik using PROPKA pH. Waters with less than 3 hydrogen bonds to non-waters were removed. The model underwent restrained minimization, converging heavy atoms to RMSD 0.30 angstroms. Docking grids suitable for peptide docking were generated using GLIDE "Receptor Grid Generation" and defining PABP1 peptide as an excluded substrate. A scaling factor 1.0 and a partial charge cutoff of 0.25 for van der waals radius scaling were used. Peptide fragments were built and minimized

locally using "3-D Builder" and all substrate arginine was aligned to co-crystal in catalytic site using "Flexible Ligand Alignment". OPLS3e force field was used to generate substrates for docking and substrates were generated for all possible states at target pH 7.0 using Epik. Stereoisomers were computed retaining specified chirality (for amino acids) and allowing for up to 10 conformers.

### In vitro proliferation assay

For proliferation assays, 10,000 cells / 200 µl medium were plated in triplicate and supplemented with increasing doses of inhibitor using 11 points with EPZ025654 or RUX (Supplementary Table 7). Cells were treated with RUX and incubated for 48 hours. In the proliferation assay with EPZ025654, cells were split 1:1 and supplemented with fresh medium and inhibitor after 2 and 4 days. Viability was measured by cellular ATP determination using the CellTiter-Glo luminescent cell viability assay (Promega) and normalized proliferation in media with an equivalent volume of DMSO. IC50 values were determined by Graph Pad Prism 8.0.

### Drug combinations and synergy analysis

Cells were seeded on 96-well plates at 40,000 cells per well and treated for 6 days with the indicated doses of EPZ025654. Fresh media and drugs were added on days 2 and 4. To evaluate synergy with RUX, cells were treated with EPZ025654 for 4 days followed by the addition of RUX for 48 hrs. The viability assay was assessed with the CellTiter Glo method on day 6. To interpret the value of synergy scores, Bills synergy was calculated using the Bioconductor package Synergy-Finder v2.0[84]. All experiments were repeated independently at two times.

### Colony formation assays in methylcellulose

Clonogenic potential of HEL and SET2 cells were plated in methylcellulose semi-solid media supplemented with DMSO (control), RUX (JAK1/JAK2-inhibitor), EPZ025654 (CARM1-inhibitor), or a combination of both (MethoCult™ H4435 Enriched, STEMCell Technologies). The concentration of RUX was used according to the IC50 values for each cell line: 500, 250, and 50 nM in HEL, UKE-1, and SET2 cells, respectively. Scoring of colonies was performed after 14 days using the automated and standardized colony counting machine STEMvision™ (STEMCell Technologies).

### Statistical analysis

The data shown were performed with at least three independent biological replicates. The results are shown as mean ± SD of three independent experiments. Data were analyzed and statistics were performed using unpaired two-tailed Student's t-test or one-way ANOVA (Graph Pad Prism 8.0). Significant differences between the two groups were noted by asterisks (*$P < 0.05$, **$P < 0.01$, ***$P < 0.001$).

## Data availability

RNA-seq and ChIP-seq data have been deposited in the NCBI Gene Expression Omnibus (GEO) database, accession code: GSE174432 and GSE200973, respectively. Source data are provided with this paper. All other remaining data are available within the article or supplementary files or are available upon request from the corresponding author S.D.N. (snimer@med.miami.edu). Source data are provided with this paper.

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

## Acknowledgements

We thank the members of the Nimer lab for their assistance and thoughtful input on the manuscript; Delphine Prou, Lauren Ashley Whitmore, and Natalia Cuevas. We also thank the Oncogenomics Shared Resource at Sylvester Comprehensive Cancer Center for RNA-sequencing services, the Biostatistics and Bioinformatics Shared Resource for data analysis, and Professor Yasushi Miyazaki (Department of Hematology, Nagasaki University Hospital, Nagasaki, Japan) for his guidance and assistance. This work was supported by funds from Sylvester Comprehensive Cancer Center, grant R01 CA251664-01 and 1P30CA240139-01 from the National Cancer Institute to S.D.N., 1F31CA254232-01 from the National Cancer Institute to A.K.M., the Translational Research Program Grant from the Leukemia and Lymphoma Society to S.D.N. and Grant-in-Aid for Scientific Research (KAKENHI) from the Japan Society for the Promotion of Science (22K08480) to H.I.

## Author contributions

H.I., S.M.G. and S.D.N. conceived the project. H.I. and S.D.N. designed the experiments and wrote the manuscript. H.I. conducted most of the experiments. A.K.M assisted with immunoprecipitation and immunoblotting experiments to evaluate CARM1 dimerization, and phospho-CARM1 immunoblotting in patient samples. A.K.M. and P-.J.H. assisted with sample preparation for mass spectrometry. P-.J.H. assisted with RUNX1 arginine methylation experiments. S.M.G. assisted with in vitro phosphorylation assay. A.K.M. and F.L. assisted with in vitro methylation assay. C.M. M.R. and J.S. contributed to the shRNA knockdown. J.S. assisted in ChiP-seq assays. D.B., C.M. and S.D. assisted immunoblotting to determine the subcellular fraction of CARM1. G.M. assisted with the library preparation for RNA-sequencing. C.C. and N.M. assisted with colony-forming assay. A.C.U. and S.S. performed the crystal structure analysis. T.T., T.B. and J.T. assisted with the collection and analysis of the patient samples.

## Competing interests

The authors declare no competing interests.

## Ethics

The study was conducted in accordance with the Declaration of Helsinki. All samples were collected from patients with informed consent according to protocols approved by the institutional review boards of Sylvester Comprehensive Cancer Center.

## Additional information

[1]Sylvester Comprehensive Cancer Center, University of Miami, Miller School of Medicine, Miami, FL 33136, USA. [2]Genomics Institute of the Novartis Research Foundation, San Diego, CA 92121, USA. [3]Department of Biochemistry and Molecular Biology, University of Miami, Miller School of Medicine, Miami, FL 33136, USA. [4]Center for Epigenetics, Memorial Sloan Kettering Cancer Center, New York, NY 10065, USA. [5]Department of Medicine, University of Miami, Miller School of Medicine, Miami, FL 33136, USA. [6]Department of Molecular and Cellular Pharmacology, University of Miami, Miller School of Medicine, Miami, FL 33136, USA. [7]Department of Medicine, Division of Hematology, Sylvester Comprehensive Cancer Center, University of Miami Health System, Miami, FL 33136, USA. ✉e-mail: snimer@med.miami.edu

