## [Peer Review File · Nature Communications]

Tyrosine phosphorylation of CARM1 promotes its enzymatic activity and alters its target specificityREVIEWER COMMENTS

Reviewer #1 (Remarks to the Author):

Summary:

In this study, the group led by Stephan Nimer found that CARM1 can be tyrosine phosphorylated by the mutated and oncogenic form of JAK2 (JAK2-V617F). This phosphorylation causes the activation of CARM1, and also subtly changes its substrate specificity. A BioID experiment identified RUNX1 as a CARM1 interacting protein. The methylation of RUNX1 by CARM1 is the focus of this study, and the authors find distinct transcriptional profiles are regulated by this axis. Importantly, small molecule inhibitors of CARM1 and JAK2 are synergistic in a subset of cell lines, raising the possibility of combination therapy.

Critique:

The manuscript is well-written and the studies are well-controlled. The key novel findings in this study are; First, CARM1 can be tyrosine phosphorylated by mutant JAK2. Second, this phosphorylation activates CARM1. Third, activated CARM1 methylates RUNX1 and alters its transcriptional activity. Fourth, therapeutic targeting of JAK2 and CARM1 together could be effective for JAK2-V617F mutant myeloid malignancies. There are a number of points that need to be addressed, before this study is suitable for publication in Nature Communications.

The following needs to be addressed:

1. The data suggesting that CARM1 tyrosine phosphorylation can regulate its ability to dimerize should be strengthened. The RUX compound seems to have a minor impact on dimerization (Fig. S6). Using the same HA/MYC tagged system, do they see an increase in CARM1 dimerization when the JAK2-V617F mutant is overexpressed.
2. They state that the data in Fig. 4D confirms that the CARM1/RUNX1 interaction is regulated by phosphorylation. This is not the case. The data indicates that these two tyrosine residues are important for the interaction, but not phosphorylation. Similar to my suggestion above, does the overexpression of JAK2-V617F induce the stabilization of the CARM1/RUNX1 interaction.
3. Over 10 years ago, the Nimer group published that PRMT5 is a substrate for JAK2-V617F. This earlier paper is very relevant to their current finding. However, this story is not even discussed or addressed in the current paper (it is referenced in the methods section).
 - a. This paper needs to be addressed in the discussion and introduction.
 - b. In the case of PRMT5, they find that JAK2-V617F mediated phosphorylation results in loss of activity. Thus, there is likely a switch from SDMA to CARM1-deposited ADMA marks. There are a number of reports that PRMT5 and CARM1 can share substrates. The authors should test if the RUNX1 R223 and R319 sites can also be substrates for PRMT5. If so, then JAK2-V617F activity may alter RUNX1 from a SDMA carrier to an ADMA carrier.

Minor points:

1. In line 44, of the abstract, a word is missing – "...in RUNX1 is lost in cells engineered to"

Reviewer #2 (Remarks to the Author):

Itonaga et al present a biochemically-dense paper linking JAK2 kinase, the arginine methyltransferase PRMT4 and the transcription factor RUNX1. The major claim is that JAK2 is a key regulator of PRMT4 to promote its dimerization, nuclear localisation and the asymmetric demethylation of RUNX1. If physiologically relevant, the findings are an important connection between growth factor signalling and epigenetic regulation of transcription. The lab has good

published experience in arginine methyltransferases. Experiments overall are well designed with appropriate controls and standard methods for protein biochemistry.

Weaknesses include the fact that most experiments performed in HEL, SET2 and UKE cells which over express mutant JAK2; lack of novelty from their previous paper showing PRMT4 can methylate RUNX1; biological consequences limited to IC50s drug combination in a few cell lines; poor explanation for the physiological function of such a pathway.

1] Can you show the pathway is intact in normal cells via unmutated JAK2? If you stimulate CD34+ cells or TF-1 with cytokine, do you see CARM1 phosphorylation or is it only observed in JAK2 V617F cell with high levels of JAK activity?

2] Can you demonstrate that peripheral blood mononuclear patient samples with JAK2 V617F mutation have high levels of CARM1 phosphorylation and RUNX1 methylation?

3] Even when mutated, JAK2 is normally bound to cytokine receptor subunits via Box 1/Box 2 with V617F causing prolonged receptor dimerization in absence of ligand. Can you immunoprecipitate a cytokine receptor complex and see CARM1 phosphorylated? Or is the JAK2 bound to CARM1 distinct from cytokine receptor associated JAK?

4] The combo data with 3 cell lines is too limited to make firm conclusions about biology or targeting. Can you show evidence that non-phosphorylatable CARM1 changes disease progression/behaviour in mouse model of JAK2 V617F? or patient samples in vitro?

5] Ruxolitinib is not selective for JAK2. Can you treat with fedratinib or more selective JAK2 inhibitor and observe loss of CARM1 phosphorylation and loss of RUNX methylation?

Minor point: line 391 "additive"

Abstract poorly written and does not reflect the best conclusions of the paper

Reviewer #3 (Remarks to the Author):

Review-Nature Communications (IF 17.69) (Itonaga et al, 2022)

Tyrosine phosphorylation of CARM1 promotes its enzymatic activity and alters its target specificity
All criteria are met for a Nature Communications article

In this manuscript Itonaga and colleagues have comprehensively investigated

The authors show elegant evidence to demonstrate JAK2 mediated tyrosine phosphorylation of CARM1 Y149 and Y334, the functional impact of this phosphorylation and how this contributes to CARM1 nuclear and chromatin localization. The role of in CARM1 and their functional impact on activity and localization. The impact of CARM1 phosphorylation on the regulation of RUNX1, and RUNX1-target genes through asymmetrical demethylation is demonstrated, where it can influence hemopoietic cell differentiation and regulation of cell cycle and survival. This study highlights CARM1 as a potential target in addition to JAK2 in JAK2-V617F driven disease, this is important research given the clinical challenges of JAK2 inhibitors.

This is a very well written, thoroughly investigated and comprehensively analysed study, the authors are to be commended.

- Figure 2 & Line 1097 - TF-1 and HEL cells are an erythroleukemic cells and SET2 cells are megakaryoblastic and have been included within the panel described to be myeloid cell lines. Please comment on their inclusion and whether the CARM1 phosphorylation observed is a myeloid specific or more general leukemia specific finding. This may have value in other JAK2 driven hematopoietic neoplasms

Minor points:

- Line 44 - Please reword "..., and the asymmetric demethylation of R223 and R319 in RUNX1 is lost in engineered to express only" as this is unclear
- Please introduce the relevance of the use of UKE cells prior to the result described in Fig S3A within the manuscript text for improved for context.
- Line 248 - describes the identification of high intensity fragments of RUNX1 interacting with CARM1-BirA* fusion in HEL and K562 cells, however no data is presented for K562 in Figure S7D.
- The material and Methods sections in the main manuscript have been directly repeated in the supplementary section, this could be removed from supplementary.
- Line 779 - rux is noted as a JAK2 inhibitor. It is a dual JAK1 and JAK2 inhibitor and it should be

noted within the manuscript.

- Please add the concentration of rux used in the legend of Figure 1A
- Figure 2C and 2E, please include the name of the cell line used in the figure
- Figure 4G and 4H, please comment on the viability of the cells following the 5/8 days exposure to rux
- Please indicate the number of assays performed for Figure 6A.
- In the figure legend for 6C the asterisk is missing for $p < 0.05$, please amend.
- Figure S2B and S2C please indicate the concentration of rux used in the figure legend.

- Figure S6B, please define WCE.
- Figure S10 – For completion it would be good to show the sequence over both the Y149 and Y334 for each line to confirm the presence of the alternate wild type Y residue in the single knock lines.
- Figure S13 – Please add the number of repeats, p-value and statistical tests done to figure legend.
- Please note the journal requires the p value to be stated in the case of non-specific findings, please amend.
- In keeping with the journal requirements references should not exceed 70.

In this study the impact of CARM1 functional regulation has been assessed in focusing on JAK2-V617F driven disease. More broadly in the future it would be interesting to assess the impact of this axis in other JAK2 driven hematological malignancies such as those containing JAK2-fusion disease where rux alone is not having a major impact on disease clinically.

Reviewer #4 (Remarks to the Author):

Itonaga and colleagues use in vitro techniques to show that activated JAK2 (V617F) can phosphorylate PRMT4/CARM1 at the highly conserved catalytic domain residues Y149 and Y334. Through cell line experiments using CARM1 mutants lacking either or both of these tyrosines they ascribe several important CARM1 functions to its tyrosine phosphorylation.

Overall the manuscript is clearly written and Figures are of high quality. As a general statement, the manuscript would benefit from a more detailed explanation/rationale for why the JAK2/CARM1 link was initially established in order to set the scene for the results.

Figure 3 data raise serious questions about the role of CARM1 phosphorylation that need to be addressed in order for the authors to substantiate their claims.

Conclusions drawn from Figure 3F dimerisation experiments are based on the assumption that transiently transfected CARM1 is being constitutively phosphorylated in 293T cells (such that the non-phosphorylatable CARM1 mutants Y149F and Y334F lose activity). Yet in Figure 3D it is clear that CARM1 phosphorylation (at least at Y334) is absent/minimal in transfected 293T cells unless JAK2 is co-transfected. Given that the dimerisation experiments were done in the absence of JAK2, how do the authors reconcile these results? Rather than supporting author claims, these results instead seem to suggest that the Y149F mutation affects CARM1 function through a mechanism unrelated to its phosphorylation.

Unless this point is addressed, it undermines several central claims of the manuscript. The non-phosphorylatable CARM1 mutants Y149F and Y334F lack H3.1 methyltransferase activity, are unable to bind RUNX1, and have reduced dimerisation. All these observations could be explained if these non-phosphorylatable mutant proteins simply fail to fold or assemble properly. This may be unrelated to the phosphorylation status of the residues in question. Can the authors rule out this alternative explanation?

This major concern extends to the cell line non-phosphorylatable CARM1 knockin mutants examined in Figure 4F and Figure 5.

The CARM1-RUNX1 interaction was previously described. Conclusions regarding CARM1 methylation of RUNX1 are well supported by loss of function experiments.

Lin3 147 (Fig 2B): The statement "confirming that JAK2 is the relevant kinase phosphorylating CARM1" is too emphatic. Changes in JAK2 activity may regulate several other kinases that could in turn directly alter CARM1 activity.

Typographical errors:

Line 44: "lost in CELLS engineered"

Line 207 "it's"

Reviewer #5 (Remarks to the Author):

The manuscript by Itonaga et al. builds upon prior work from the same group evaluating the role of the enzyme CARM1 in the development of hematologic malignancies. Here, the authors specifically turn to understanding the potential mechanisms regulating CARM1 activity. The major conclusion of the authors' work is that JAK2 phosphorylates CARM1 at its active site, directly impacting its enzymatic activity and leukemogenic function.

Overall, the data in Fig. 3 and beyond is convincing that CARM1 phosphorylation impacts its enzymatic activity, which I believe is an important advance in the field. In general, the authors are commended for integrating solid molecular biology as well as BioID approaches into the work. However, based on technical aspects of the JAK2-related experiments, it is not yet clear whether JAK2 is primarily responsible for this CARM1 phosphorylation, or whether the link to JAK2 is an artifact of the experimental designs used and another kinase is the main effector in this context. This link between JAK2 and CARM1 remains the weakest component of the proposed mechanism in an otherwise interesting work.

Major:

1. The authors begin the manuscript with a curiously specific hypothesis: that JAK2 phosphorylates CARM1. I agree the recombinant protein and mass spectrometry data certainly support that JAK2 can drive phosphorylation, at least in this reconstituted *in vitro* system. However, it is surprising that the authors would go through this initial effort to probe this relationship without some additional, earlier, data to suggest a meaningful interaction between these proteins. Building evidence of a physiological link between these proteins (whether bioinformatic, clinical, or otherwise), prior to *in vitro* protein validation, would strengthen the initial rationale.
2. Fig S2E: it is unclear whether the raised polyclonal antibodies are particularly specific for the noted CARM1 phospho-peptides. From this analysis, it appears that many of the noted bands in whole cell lysates are decreased after Rux treatment; this phenotype is not selective for the purported CARM1 band. Also, there is not any convincing evidence that the highlighted band in the whole cell lysate is truly phospho-CARM1; for example, the total CARM1 antibody appears to have a doublet band, why is this not seen for the phospho antibodies?
3. Related to the prior comment, this uncertainty about what the antibody is truly staining brings significant questions regarding the reliability of the Data in Fig. 2A. For example, there appears to be CARM1 p-Tyr344 signal in normal CD34+ cells, but there is no signal for total CARM1; this finding raises suspicion that the CARM1 p-Tyr334 signal is a non-specific background band. Therefore, the conclusion that CARM1 phosphorylation correlates with JAK2 mutation status is not as strong as it could be. In addition, uncertainty about the antibody performance affects the interpretation of Fig. 2F and Fig. 3G as well (for example, in 3G, the interpretation of these subcellular localization blots would be entirely different if the purported pTyr signal is actually just a background band). Though technically challenging, confidence would be greatly bolstered in these results if the authors could use an orthogonal technology – perhaps pulldown followed by targeted mass spectrometry, guided by their MS spectra in Fig. 1A – to show that the intensity of antibody signal in whole cell lysates correlates with the intensity of MS peptide signal for the phosphorylated peptide.

4. The authors mention "data not shown" when evaluating binding of CARM1-HA to JAK1, JAK3, or TYK2. This data should be included in the paper to demonstrate at least some degree of specificity for JAK2.
5. Analogous to experiments in Fig 4E, the authors should perform JAK2 knockdown and evaluate alterations of CARM1 phosphorylation. The experiments in Fig. 4G are relatively circumstantial given that ruxolitinib treatment may be leading to indirect changes in substrate methylation via impacts on other JAK-downstream pathways, and not necessarily functioning through CARM1.
6. The immunoprecipitation experiments showing binding of JAK2 to CARM1-HA are all done in the context of CARM1 overexpression. Is there any evidence that this interaction can be detected with endogenous expression levels of CARM1? Given the presented data, some concern remains that the noted interaction is an artifact of the overexpression system. This concern remains given the lack of any JAK2-CARM1 interaction found in the BioID experiments.
7. Regarding BioID experiments, The authors write "we identified 128 and 60 proteins that significantly interact with CARM1 in HEL and K562 cells." However, in the figure and the supplementary dataset, I could not find any statistical test that was specifically used to determine this cutoff of interactors vs. non-interactors. This should be clarified, as the nature of this significance cutoff impacts the downstream network analysis used.
8. The synergy impacts of the EPZ molecule and ruxolitinib remain relatively unconvincing, at least related to the proposed JAK2-CARM1 axis here. Can the authors provide more evidence that the noted synergy is related to their proposed mechanism? It is easy to imagine this synergy could result from other cellular impacts of these two agents.

Minor

1. In the mass spectrum figure panels (Fig. 1B), it would be preferable to alter the y-axis scale so that the intensity of the noted fragment ions do not overlap the amino acid labels.
2. Based on the mass spectrum shown in Fig S5B, it is not clear whether any fragment ions specifically denote and localize the two proposed methylation sites in this peptide.
3. Discussion line 460: does the CRISPR screen data in DepMap match that from the DepMap shRNA screens (as referenced in the manuscript), regarding CARM-1 essentiality? In general the CRISPR screen data in DepMap is seen to be more reliable than the shRNA.
4. Fig. S7D: what type of targeted mass spectrometry approach is used here? Is this parallel reaction monitoring (this appears to be the case based on the Methods)? This approach should be specified in the main text and/or figure legend.

Reviewer #1 (Remarks to the Author):

Summary:

In this study, the group led by Stephen Nimer found that CARM1 can be tyrosine phosphorylated by the mutated and oncogenic form of JAK2 (JAK2-V617F). This phosphorylation causes the activation of CARM1, and also subtly changes its substrate specificity. A BioID experiment identified RUNX1 as a CARM1 interacting protein. The methylation of RUNX1 by CARM1 is the focus of this study, and the authors find distinct transcriptional profiles are regulated by this axis. Importantly, small molecule inhibitors of CARM1 and JAK2 are synergistic in a subset of cell lines, raising the possibility of combination therapy.

Critique:

The manuscript is well-written and the studies are well-controlled. The key novel findings in this study are; First, CARM1 can be tyrosine phosphorylated by mutant JAK2. Second, this phosphorylation activates CARM1. Third, activated CARM1 methylates RUNX1 and alters its transcriptional activity. Fourth, therapeutic targeting of JAK2 and CARM1 together could be effective for JAK2-V617F mutant myeloid malignancies. There are a number of points that need to be addressed, before this study is suitable for publication in Nature Communications.

Response: We thank the reviewer for concluding that the findings of our study are novel and well described in our manuscript. We appreciate the careful review of our work and have addressed all comments in this point-by-point response and in our revised manuscript. We hope this carefully revised manuscript will be deemed suitable for publication in Nature Communications.

The following needs to be addressed:

1. The data suggesting that CARM1 tyrosine phosphorylation can regulate its ability to dimerize should be strengthened. The RUX compound seems to have a minor impact on dimerization (Fig. S6). Using the same HA/MYC tagged system, do they see an increase in CARM1 dimerization when the JAK2-V617F mutant is overexpressed.

Response: As requested, we co-transfected HA-tagged CARM1, MYC-tagged CARM1, and JAK2 (WT or V617F mutants) constructs into 293T cells, immunoprecipitated with anti-HA antibodies, and probed with anti-MYC antibodies to further characterize CARM1 dimerization. The signal for MYC-CARM1 was higher in 293T cells expressing JAK2-V617F mutant than in 293T cells expressing WT-JAK2, supporting our conclusion about

the impact of CARM1 phosphorylation on its dimerization. We added this result as supplemental Figure S6B.

2. They state that the data in Fig. 4D confirms that the CARM1/RUNX1 interaction is regulated by phosphorylation. This is not the case. The data indicates that these two tyrosine residues are important for the interaction, but not phosphorylation. Similar to my suggestion above, does the overexpression of JAK2-V617F induce the stabilization of the CARM1/RUNX1 interaction.

Response: To address this concern, we did conduct experiments where we expressed WT-JAK2 or JAK2-V617F in K562 cells but did not see an increase in CARM1 phosphorylation (at Y149 or Y334) following the overexpression of the wild-type or the mutant, constitutively active kinase, as shown in the left figure below. The reason for this is found in the right figure, as neither WT-JAK2 nor V617F-JAK2 is phosphorylated in K562 cells. Thus, we believe that hyperactive phosphorylated JAK2, via genetic mutation and cross phosphorylation, is needed to trigger for the maximal ability to phosphorylate CARM1 on Y149 and Y334 in hematopoietic cells.

3. Over 10 years ago, the Nimer group published that PRMT5 is a substrate for JAK2-V617F. This earlier paper is very relevant to their current finding. However, this story is not even discussed or addressed in the current paper (it is referenced in the methods section).

a. This paper needs to be addressed in the discussion and introduction.

Response: We appreciate this reviewer's comments about our previous study, and have added descriptions of our prior findings in both the introduction (page 7, line 108-111) and the discussion sections (page 30, line 508-511).

b. In the case of PRMT5, they find that JAK2-V617F mediated phosphorylation results in loss of activity. Thus, there is likely a switch from SDMA to CARM1-deposited ADMA marks. There are a number of reports that PRMT5 and CARM1 can share substrates. The authors should test if the RUNX1 R223 and R319 sites can also be substrates for PRMT5. If so, then JAK2-V617F activity may alter RUNX1 from a SDMA carrier to a ADMA carrier.

Response: We thank the reviewer for raising this question. To address this issue, we examined the methyltransferase activity of PRMT5 on RUNX1 in Figure S8A. We did not observe methylation of RUNX1-R223 or -R319 upon incubation with PRMT5, based on a standard in vitro methylation assay.

Minor points:

1. In line 44, of the abstract, a word is missing – “...in RUNX1 is lost in cells engineered to”

Request: We thank the reviewer’s comments. We have revised the description: “While non-phosphorylatable CARM1 methylates some substrates (e.g. BAF155 and PABP1), only phospho-CARM1 methylates the RUNX1 transcription factor, on R223 and R319.”

Reviewer #2 (Remarks to the Author):

Itonaga et al present a biochemically-dense paper linking JAK2 kinase, the arginine methyltransferase PRMT4 and the transcription factor RUNX1. The major claim is that JAK2 is a key regulator of PRMT4 to promote its dimerization, nuclear localisation and the asymmetric demethylation of RUNX1. If physiologically relevant, the findings are an important connection between growth factor signalling and epigenetic regulation of transcription. The lab has good published experience in arginine methyltransferases. Experiments overall are well designed with appropriate controls and standard methods for protein biochemistry.

Weaknesses include the fact that most experiments performed in HEL, SET2 and UKE cells which over express mutant JAK2; lack of novelty from their previous paper showing PRMT4 can methylate RUNX1; biological consequences limited to IC50s drug combination in a few cell lines; poor explanation for the physiological function of such a pathway.

Response: We thank the reviewer for mentioning that the experiments were well designed with appropriate controls. We also appreciate the constructive comments about the manuscript's shortcomings, which we have addressed below and in the revised version including adding our analysis of samples from patients with myeloid neoplasms to demonstrate the physiologic relevance of our findings.

1. Can you show the pathway is intact in normal cells via unmutated JAK2? If you stimulate CD34+ cells or TF-1 with cytokine, do you see CARM1 phosphorylation or is it only observed in JAK2 V617F cell with high levels of JAK activity?

Response: As shown in Figure 2A, CD34-positive cord blood cells (normal hematopoietic stem cells [HSC]) showed relatively low levels of CARM1 protein. Based on this fact, and because it is difficult to induce CARM1 phosphorylation by cytokine stimulation, we did not check the CARM1 phosphorylation in normal HSC. However, we did use TF-1 cells, as suggested, for these experiments. We were not to increase CARM1 phosphorylation by stimulating TF-1 cells, an erythroleukemia cell line expressing the erythropoietin receptor (Blood. 1988;71:104-109.) with cytokines (as shown for EPO, in the below figures).

After 20 hours of serum/growth factor starvation, TF-1 cells were treated with 10U/mL of erythropoietin (EPO). We observed JAK and STAT3 phosphorylation in these cells but did not see any change in the level of phospho-CARM1. These experiments were repeated, generating two independent analyses, and both yielded the same result.

2. Can you demonstrate that peripheral blood mononuclear patient samples with JAK2 V617F mutation have high levels of CARM1 phosphorylation and RUNX1 methylation?

Response: Per the reviewer's suggestion, we evaluated CARM1 phosphorylation in samples from patients with myeloid neoplasms. We observed CARM1 phosphorylation in samples from patients with JAK2-V617F mutation-positive myeloproliferative neoplasms but not patients with JAK2-WT acute myeloid leukemia or chronic myeloid leukemia (page 12, line 196-page 13, line 203; and Figure 2G). These results confirmed the presence of a phospho-JAK2-CARM1 axis and implied that phospho-CARM1 is primarily found in cells with highly activated JAK2V617F (addressing given #1 of reviewer 2). We have added the information about these patients, in the supplemental experimental procedures and in Table S2.

3. Even when mutated, JAK2 is normally bound to cytokine receptor subunits via Box 1/Box 2 with V617F causing prolonged receptor dimerization in absence of ligand. Can you immunoprecipitate a cytokine receptor complex and see CARM1 phosphorylated? Or is the JAK2 bound to CARM1 distinct from cytokine receptor associated JAK?

Response: We could not find any cytokine receptors interacting with CARM1 in the BioID assay (now Figure S7D). As shown in Figure S3C, immunoprecipitations did not show binding of CARM1 to JAK1, JAK3, or TYK2, which are bound to cytokine receptors in JAK2-V617F mutant cells (Cancer Cell. 2015;28:15-28.). In addition, based on the above results in TF-1 cells treated with EPO, activation of the EPO receptor does not directly

induce CARM1 phosphorylation. These facts suggest that CARM1 does not directly bind cytokine receptors.

4. The combo data with 3 cell lines is too limited to make firm conclusions about biology or targeting. Can you show evidence that non-phosphorylatable CARM1 changes disease progression/behaviour in mouse model of JAK2 V617F? or patient samples in vitro?

Response: In response to this comment, we have modulated our statements about biology or targeting. To fully address this suggestion, we would have to create genetically modified mice that express non-phosphorylatable CARM1 which is beyond the scope of this manuscript. We are not sure how to address this issue using patient samples. As shown in Figure 2G, we did examine CARM1 phosphorylation and JAK2 phosphorylation in samples from patients with JAK2-V617F mutation-positive myeloproliferative neoplasms and observed more phospho-CARM1 in samples with greater phospho-JAK2 (lane 4) than in samples with less phospho-JAK2 (lanes 5 and 6), confirming the relationship between JAK2 phosphorylation and CARM1 phosphorylation in patient samples. Increasing JAK2 phosphorylation is associated with disease progression in patients with JAK2-V617F mutation-positive myeloproliferative neoplasms (Cancer Cell. 2015;28:15-28.). Thus, further studies can clarify the clinical implications of CARM1 phosphorylation in disease progression.

5. Ruxolitinib is not selective for JAK2. Can you treat with fedratinib or more selective JAK2 inhibitor and observe loss of CARM1 phosphorylation and loss of RUNX methylation?

Response: To address this concern, we used NVP-BSK805, highly selective JAK2 inhibitors (Mol Cancer Ther. 2010;9:1945-1955.) rather than fedratinib (TG101348) which has potential inhibitory effect against JAK1 and FLT3 (Cancer Cell. 2008;13:311-320.). We did observe that NVP-BSK895 decreased CARM1-Y334 phosphorylation although its effect appeared to be weaker than that of RUX (Figure S2F). These results suggest that the inhibition of both JAK1 and JAK2 may be more important to decrease CARM1 phosphorylation than inhibition of JAK2 alone.

Minor point: line 391 “additive”

Response: We have revised this word, on page 25, line 428.

6. Abstract poorly written and does not reflect the best conclusions of the paper.

Response: As suggested, we have revised the abstract to better present our conclusions and make our findings easier to understand by the readers.

Reviewer #3 (Remarks to the Author):

Review-Nature Communications (IF 17.69) (Itonaga et al, 2022)

Tyrosine phosphorylation of CARM1 promotes its enzymatic activity and alters its target specificity

All criteria are met for a Nature Communications article.

In this manuscript Itonaga and colleagues have comprehensively investigated.

The authors show elegant evidence to demonstrate JAK2 mediated tyrosine phosphorylation of CARM1-Y149 and -Y334, the functional impact of this phosphorylation and how this contributes to CARM1 nuclear and chromatin localization. The role of in CARM1 and their functional impact on activity and localization. The impact of CARM1 phosphorylation on the regulation of RUNX1, and RUNX1-target genes through asymmetrical demethylation is demonstrated, where it can influence hemopoietic cell differentiation and regulation of cell cycle and survival. This study highlights CARM1 as a potential target in addition to JAK2 in JAK2-V617F driven disease, this is important research given the clinical challenges of JAK2 inhibitors.

This is a very well written, thoroughly investigated and comprehensively analysed study, the authors are to be commended.

Response: We thank the reviewer for concluding that our work thoroughly investigated and comprehensively analyzed CARM1 phosphorylation and that our manuscript was well written. We also appreciate the reviewer's suggestions and have revised the manuscript to respond to them.

1. Figure 2 & Line 1097 - TF-1 and HEL cells are an erythroleukemic cells and SET2 cells are megakaryoblastic and have been included within the panel described to be myeloid cell lines. Please comment on their inclusion and whether the CARM1 phosphorylation observed is a myeloid specific or more general leukemia specific finding. This may have value in other JAK2 driven hematopoietic neoplasms.

Response: We thank the reviewer for this question and now explain the rationale better. According to the WHO classification, acute myeloid leukemia includes acute monocytic leukemia (e.g. MOLM13), acute erythroid leukemia (e.g. HEL and UKE-1 cells), and acute megakaryoblastic leukemia (e.g. SET-2 cells). We selected the panel of myeloid leukemia cell lines according to our previous studies as they are representative of different "myeloid lineages" (S. Greenblatt, et al. Cancer Cell. 2018;33:1111-1127.). As shown in Figure 2G, CARM1 phosphorylation was observed in patients with primary

myelofibrosis, polycythemia vera, and essential thrombocythemia. Considering that the JAK2-V617F mutation is detected among patients with myelodysplastic syndromes, acute myeloid leukemia, and myeloproliferative neoplasms, including chronic myelomonocytic leukemia (page 7, line 99-106), it appears that CARM1 phosphorylation is found in various types of myeloid neoplasms. We describe the potential relationship between disease type/stage and CARM1 phosphorylation in the discussion section (page 30, line 520-521).

Minor points:

2. Line 44 - Please reword "..., and the asymmetric demethylation of R223 and R319 in RUNX1 is lost in engineered to express only ..." as this is unclear.

Response: We provide a clearer description to explain that phospho-CARM1 asymmetrically dimethylates RUNX1.

3. Please introduce the relevance of the use of UKE cells prior to the result described in Fig S3A within the manuscript text for improved for context.

Response: As suggested, we described the reason for using UKE cells (page 10, line 174-176) and added the appropriate reference (Blood Cancer J. 2012;2(4):e66.).

4. Line 248 - describes the identification of high intensity fragments of RUNX1 interacting with CARM1-BirA* fusion in HEL and K562 cells, however no data is presented for K562 in Figure S7D.

Response: We are sorry that these descriptions were misleading. We performed targeted mass spectrometry analysis to confirm the results of shotgun mass spectrometry conducted using HEL cells expressing CARM1-BirA* or BirA* alone. Because an interaction between CARM1 and RUNX1 was not observed in K562 cells, we did not perform targeted mass spectrometry. We have revised the description of these experiments on page 16, line 267.

5. The material and Methods sections in the main manuscript have been directly repeated in the supplementary section, this could be removed from supplementary.

Response: Considering the word limitation in the main manuscript, we now only provide an extensive description in the supplemental material and methods section.

6. Line 779 - rux is noted as a JAK2 inhibitor. It is a dual JAK1 and JAK2 inhibitor and it should be noted within the manuscript.

Response: We thank the reviewer for pointing this out. We have corrected our description of RUX from “JAK2 inhibitor” to “JAK1/JAK2 inhibitor”.

7. Please add the concentration of rux used in the legend of Figure 1A.

Response: We have added the concentration of RUX used to the legend of Figure 1A (page 45, line 768).

8. Figure 2C and 2E, please include the name of the cell line used in the figure.

Response: We have added the names of the cell lines used to these figures.

9. Figure 4G and 4H, please comment on the viability of the cells following the 5/8 days exposure to rux.

Response: We added the data on cell viability in the figure legends for Figure 4G and 4H (which is now Figure 4E and 4F)(page 52, line 889-890; and page 52, line 893-895).

10. Please indicate the number of assays performed for Figure 6A.

Response: We now indicate the number of assays performed for Figure 6A (page 54, line 937-938).

11. In the figure legend for 6C the asterisk is missing for $p < 0.05$, please amend.

Response: We have increased the size of the asterisk in Figure 6C, so it is visible.

12. Figure S2B and S2C please indicate the concentration of rux used in the figure legend.

Response: We have added the concentration of RUX to the Figure S2B legend (we did not use RUX in Figure S2C).

13. Figure S6B, please define WCE.

Response: We now define “WCE” in Figure S6B (which is now Figure S6C in the revised manuscript)(whole cell extract).

14. Figure S10 – For completion it would be good to show the sequence over both the Y149 and Y334 for each line to confirm the presence of the alternate wild type Y residue in the single knock lines.

Response: As your suggestion, we modified Figure S10A to show the sequence data around Y149 and Y334.

15. Figure S13 – Please add the number of repeats, p-value and statistical tests done to figure legend.

Response: We have added the number of replicate experiments, statistical tests, and p-values to Figure S13.

16. Please note the journal requires the p value to be stated in the case of non-specific findings, please amend.

Response: As suggested, we have revised the text and figures in Figure 4G (page 52, line 894-895) and Figure S12.

17. In keeping with the journal requirements references should not exceed 70.

Response: We thank the reviewer and have removed the less important references to meet the journal requirements.

In this study the impact of CARM1 functional regulation has been assessed in focusing on JAK2-V617F driven disease. More broadly in the future it would be interesting to assess the impact of this axis in other JAK2 driven hematological malignancies such as those containing JAK2-fusion disease where rux alone is not having a major impact on disease clinically.

Response: We agree with the reviewer that this suggestion is quite interesting. *JAK2*-rearrangement is observed in myeloid/lymphoid neoplasms with eosinophilia and tyrosine kinase gene fusions (Leukemia. 2022;36(7):1703-1719.) and Ph-like acute lymphoblastic leukemia (Cancer Cell. 2012;22:153-166.)(Blood. 2012;120:3510-3518.). However, these diseases are rare clinical entities and cell lines that express JAK2 fusions are not commercially available. For these reasons, we were not able to examine the role of CARM1 phosphorylation in *JAK2* fusion driven hematological malignancies in this study.

Reviewer #4 (Remarks to the Author):

Itonaga and colleagues use in vitro techniques to show that activated JAK2 (V617F) can phosphorylate PRMT4/CARM1 at the highly conserved catalytic domain residues Y149 and Y334. Through cell line experiments using CARM1 mutants lacking either or both of these tyrosines they ascribe several important CARM1 functions to its tyrosine phosphorylation.

Overall the manuscript is clearly written and Figures are of high quality. As a general statement, the manuscript would benefit from a more detailed explanation/rationale for why the JAK2/CARM1 link was initially established in order to set the scene for the results.

Response: We appreciate the reviewer's comments and provide more background information that describe the underlying findings and hypothesis that led to these studies, in particular the differential sensitivity of AML cell lines to CARM1 inhibition.

1. Figure 3 data raise serious questions about the role of CARM1 phosphorylation that need to be addressed in order for the authors to substantiate their claims.

Conclusions drawn from Figure 3F dimerisation experiments are based on the assumption that transiently transfected CARM1 is being constitutively phosphorylated in 293T cells (such that the non-phosphorylatable CARM1 mutants Y149F and Y334F lose activity). Yet in Figure 3D it is clear that CARM1 phosphorylation (at least at Y334) is absent/minimal in transfected 293T cells unless JAK2 is co-transfected. Given that the dimerisation experiments were done in the absence of JAK2, how do the authors reconcile these results? Rather than supporting author claims, these results instead seem to suggest that the Y149F mutation affects CARM1 function through a mechanism unrelated to its phosphorylation.

Response: We thank the reviewer for pointing these concerns out to us, which probably resulted from an incomplete description of the methods used for the experiments shown in Figure 3. In Figure 3D, we used recombinant CARM1 proteins produced in *E.coli* (HMT-11-120, Reaction Biology). In Figure 3C and 3F, we used WT or non-phosphorylatable mutant CARM1 proteins using the immunoprecipitants from 293T cells. We added the descriptions in the legends of figures to provide clearer understanding for the readers (page 48, line 831-832 and page 49, line 839-841). Furthermore, we co-transfected HA-tagged CARM1, MYC-tagged CARM1, and JAK2 (WT and V617F mutants) constructs into 293T cells, immunoprecipitated with anti-HA antibodies, and probed with anti-MYC antibodies. The signals of MYC-CARM1 were higher in 293T cells expressing JAK2-V617F mutant than those expressing WT-JAK2, supporting our

conclusion about the impact of CARM1 phosphorylation on its dimerization. We added this result as Figure S6B.

2. Unless this point is addressed, it undermines several central claims of the manuscript. The non-phosphorylatable CARM1 mutants Y149F and Y334F lack H3.1 methyltransferase activity, are unable to bind RUNX1, and have reduced dimerisation. All these observations could be explained if these non-phosphorylatable mutant proteins simply fail to fold or assemble properly. This may be unrelated to the phosphorylation status of the residues in question. Can the authors rule out this alternative explanation?

Response: We have previously reported the effects of CARM1/PRMT4 depletion and genetic deletion (S. Greenblatt, et al. *Cancer Cell*. 2018;33:1111-1127.); and the effects of an E267Q (EQ mutant) form of CARM1 that lacks methyltransferase activity, indicating that these Y to F mutant proteins are not loss of the function (LOF) mutations. As shown in Figure 4F, the single Y to F CARM1 mutants, and the double mutant CARM1 protein do methylate BAF155 and PABP1; they retain activity, thus their structure must be intact. In contrast, we previously showed that hematopoietic cells expressing an enzymatical dead mutant CARM1-EQ mutant completely lost asymmetrical dimethylation of BAF155 and PABP1. The Y to F mutants are not enzymatically dead. We wish to address the concern of reviewer #4 about the non-phosphorylatable CARM1 mutants. In addition, we have already reported the enzymatic activity and biological impacts of enzymatic dead CARM1 (S. Greenblatt, et al. *Cancer Cell*. 2018;33:1111-1127.), which was different from non-phosphorylatable CARM1 (Y to F mutants). We added the descriptions and figures to provide clearer interpretations with the results of our previous studies (page 19, line 309-317 and page 20, line 342- page 21, line 344).

3. This major concern extends to the cell line non-phosphorylatable CARM1 knockin mutants examined in Figure 4F and Figure 5.

Response: We believe that the data presented in our manuscript, and in our previous report on CARM1 depletion and the *Carm1* knockout mouse (S. Greenblatt, et al. *Cancer Cell*. 2018;33:1111-1127.), indicate that these Y to F mutant proteins are not LOF mutations. The experiments in our previous study also demonstrated effects on cell cycle, apoptosis, and differentiation, that resemble but are more extensive than the effects seen by knocking in Y to F amino acid substitutions into the CARM1 gene in the HEL cell line. Taken together these studies demonstrate that WT CARM1 plays a role in blocking differentiation and preventing the apoptosis of AML cells. The effect of the Y to F mutations on cell cycle are different than depletion of CARM1; depletion leads to a

relative G1/S hang-up, while the point mutations lead to a G2/M relative arrest. We have now added a sentence that address this point as well as the issue raised in point #2 (page 19, line 309-317 and page 20, line 342- page 21, line 344).

4. The CARM1-RUNX1 interaction was previously described. Conclusions regarding CARM1 methylation of RUNX1 are well supported by loss of function experiments.

Response: We agree that our previous work documented the RUNX1-CARM1 interaction and the methylation of RUNX1 on R223. This work identified another site of RUNX1 methylation and showed the importance of CARM1 phosphorylation for the arginine methylation of RUNX1. Our early work showed that an enzymatically dead form of CARM1 lost the dimethylation of RUNX1-R223 (Cell Rep. 2013;5:1625-1638.). To provide a clearer understanding for the readers, we added explanations about these data (page 19, line 309-317).

5. Lin3 147 (Fig 2B): The statement “confirming that JAK2 is the relevant kinase phosphorylating CARM1” is too emphatic. Changes in JAK2 activity may regulate several other kinases that could in turn directly alter CARM1 activity.

Response: As suggested, we have modified this statement to allow for the other kinases that participate in the phosphorylation of CARM1 (page 10, line 159-160).

Reviewer #5 (Remarks to the Author):

The manuscript by Itonaga et al. builds upon prior work from the same group evaluating the role of the enzyme CARM1 in the development of hematologic malignancies. Here, the authors specifically turn to understanding the potential mechanisms regulating CARM1 activity. The major conclusion of the authors' work is that JAK2 phosphorylates CARM1 at its active site, directly impacting its enzymatic activity and leukemogenic function.

Overall, the data in Fig. 3 and beyond is convincing that CARM1 phosphorylation impacts its enzymatic activity, which I believe is an important advance in the field. In general, the authors are commended for integrating solid molecular biology as well as BioID approaches into the work. However, based on technical aspects of the JAK2-related experiments, it is not year clear whether JAK2 is primarily responsible for this CARM1 phosphorylation, or whether the link to JAK2 is an artifact of the experimental designs used and another kinase is the main effector in this context. This link between JAK2 and CARM1 remains the weakest component of the proposed mechanism in an otherwise interesting work.

Response: We appreciate the reviewer's comments and have modified the text to reflect the possibility that other kinases may play a role in CARM1 phosphorylation. We do think that we had not clearly delineated that we were using bacterially expressed CARM1 in Figure 3D but mammalian cell expressed CARM1 in other experiments shown in this Figure. We first added the detailed descriptions to show why we focused on the relationship between JAK2 and CARM1. In addition, it is difficult to use JAK2-knockdown cell lines harboring JAK2 mutation because mutated JAK2 has a fundamental role as a driver gene alteration. Therefore, we showed the reduced phosphorylation of CARM1 in cells treated with not only JAK1/JAK2 inhibitor (ruxolitinib) but also highly selective JAK2 inhibitor (NVP-BSK856)(Figure S2F). Furthermore, transactivation of JAK2 by JAK1 and TYK2 has been known as alternative pathway during JAK2 inhibition (Nature. 2012;489:155-159.)(Cancer Cell. 2015;28:15-28.). In general, activation of another kinase would likely be mutually exclusive in JAK2-V617F mutant cells (Blood. 2019;134:199-210.). Therefore, we showed the reduced phosphorylation of CARM1 in JAK1/TYK2-knockout HEL cells (Figure 2F). Finally, we revised the graphical manuscript to present better understanding of the JAK2-CARM1 axis for the readers. We believe the revised manuscript could provide the clearer findings into the role of CARM1 in myeloid neoplasms harboring hyperactivated JAK2.

Major:

1. The authors begin the manuscript with a curiously specific hypothesis: that JAK2 phosphorylates CARM1. I agree the recombinant protein and mass spectrometry data certainly support that JAK2 can drive phosphorylation, at least in this reconstituted in vitro system. However, it is surprising that the authors would go through this initial effort to probe this relationship without some additional, earlier, data to suggest a meaningful interaction between these proteins. Building evidence of a physiological link between these proteins (whether bioinformatic, clinical, or otherwise), prior to in vitro protein validation, would strengthen the initial rationale.

Response: We thank the reviewer for requesting that we better explain the rationale for conducting these studies. Based on our previous work showing that JAK2 phosphorylates PRMT5 and alter its enzymatic activity (F. Liu, et al. *Cancer Cell*. 2011;19:283-294.), we embarked on these studies when we examined the sensitivity of AML cell lines to CARM1 inhibition, and noticed that JAK2-V617F mutation-positive cell lines were less sensitive to selective CARM1 inhibition (S. Greenblatt, et al. *Cancer Cell*. 2018;33:1111-1127.). We examined whether JAK2 altered CARM1 activity and found the positive link between CARM1 phosphorylation and its methylation activity. The rationale is now described in detail in our revised manuscript (page 6, line 88-90 and page 7, line 107-page 8, line 111).

2. Fig S2E: it is unclear whether the raised polyclonal antibodies are particularly specific for the noted CARM1 phospho-peptides. From this analysis, it appears that many of the noted bands in whole cell lysates are decreased after Rux treatment; this phenotype is not selective for the purported CARM1 band. Also, there is not any convincing evidence that the highlighted band in the whole cell lysate is truly phospho-CARM1; for example, the total CARM1 antibody appears to have a doublet band, why is this not seen for the phospho antibodies?

Response: As reported in the previous studies, CARM1 has two alternative splice isoforms, full-length CARM1 and an isoform lacking exon15 (CARM1 Δ E15) (*J Biol Chem*. 2005;280:28927-28935.)(*PLoS One*. 2015;10:e0128143.). CARM1 Δ E15 shows tissue-specific expression patterns. We have also shown the various expression patterns of full-length CARM1 and CARM1 Δ E15 in the leukemia cell lines (S. Greenblatt, et al. *Cancer Cell*. 2018;33:1111-1127.), and that the antibody against total CARM1 identify both full-length CAMR1 and CARM1 Δ E15. Based on the in vitro kinase assay demonstration that JAK2 can phosphorylate full-length CARM1, we believe that the

immunoblotting, using lysates from 4 AML cell lines, identified mainly the phosphorylation of full-length CARM1.

We also wish to address the concern of reviewer #5 about the specificity of the phospho-specific antibodies that we generated and used. We used Y to F (non-phosphorylatable) mutant forms of MYC-tagged CARM1 in an *in vitro* kinase assay and showed that each antibody specifically detects the phosphorylation of Y149 or Y334 (Figure S2D).

3. Related to the prior comment, this uncertainty about what the antibody is truly staining brings significant questions regarding the reliability of the Data in Fig. 2A. For example, there appears to be CARM1 p-Tyr344 signal in normal CD34+ cells, but there is no signal for total CARM1; this finding raises suspicion that the CARM1 p-Tyr344 signal is a non-specific background band. Therefore, the conclusion that CARM1 phosphorylation correlates with JAK2 mutation status is not as strong as it could be. In addition, uncertainty about the antibody performance affects the interpretation of Fig. 2F and Fig. 3G as well (for example, in 3G, the interpretation of these subcellular localization blots would be entirely different if the purported pTyr signal is actually just a background band). Though technically challenging, confidence would be greatly bolstered in these results if the authors could use an orthogonal technology – perhaps pulldown followed by targeted mass spectrometry, guided by their MS spectra in Fig. 1A – to show that the intensity of antibody signal in whole cell lysates correlates with the intensity of MS peptide signal for the phosphorylated peptide.

Response: We thank the reviewer for this question but point out that such study using MS analysis are very difficult to do because of small volume of phospho-CARM1 from the nucleus and the chromatin fraction.

As shown in Figure 2A, the antibody against CARM1-pY149 showed less background band. To correctly interpret the impact of CARM1 phosphorylation on its localization as the reviewer mentioned, we have evaluated the localization of phospho-CARM1 using antibodies against not only CARM1-pY334 but also CARM1-pY149 in Figure 3G.

Moreover, to further evaluate the effect of CARM1 phosphorylation on its localization, we performed immunoblotting with antibody against total CARM1 (neither CARM1-pY149 nor -pY334) in HEL cells after RUX treatment. Immunoblotting assay revealed that RUX decreased the proportion of CARM1 in the nucleus and in the chromatin fraction, indicating that CARM1 phosphorylation affects the subcellular localization CARM1 (Figure S6D).

4. The authors mention “data not shown” when evaluating binding of CARM1-HA to JAK1,

JAK3, or TYK2. This data should be included in the paper to demonstrate at least some degree of specificity for JAK2.

Response: As suggested by reviewer #5, we now show the data that demonstrate the absence of CARM1 binding with JAK1, JAK3, or TYK2, in Figure S3C.

5. Analogous to experiments in Fig 4E, the authors should perform JAK2 knockdown and evaluate alterations of CARM1 phosphorylation. The experiments in Fig. 4G are relatively circumstantial given that ruxolitinib treatment may be leading to indirect changes in substrate methylation via impacts on other JAK-downstream pathways, and not necessarily functioning through CARM1.

Response: To address this concern of reviewer #5, we would point out, that HEL, SET2, and UKE-1 cell lines were derived from JAK2-V617F mutant driven myeloid neoplasm cells. In the other words, the JAK2-V617F mutation is the fundamental gene alteration in these cell lines. JAK2-knockdown does have a fatal effect on these cell lines, making it difficult to accurately evaluate the direct effects of JAK2-knockdown on CARM1 phosphorylation. The use of tyrosine kinase inhibitors is always fraught with compensatory effect. The time course of methylation and re-methylation of different substrates shows that multiple pathways always brought out from initiating events.

6. The immunoprecipitation experiments showing binding of JAK2 to CARM1-HA are all done in the context of CARM1 overexpression. Is there any evidence that this interaction can be detected with endogenous expression levels of CARM1? Given the presented data, some concern remains that the noted interaction is an artifact of the overexpression system. This concern remains given the lack of any JAK2-CARM1 interaction found in the BioID experiments.

Response: We and others have previously shown the hit-and-run nature of contain enzyme-substrate interaction which may be why we do not see the binding of CARM1 to JAK2 in co-IPs using HEL cells. The original Bio-ID study did not identify the interaction between kinase and many of its substrates (Nat Commun. 2018;9:1188.), and the most likely reason is that JAK2 is weakly and/or only transiently associated with CARM1 (shown in Figure legend of Figure S7D).

7. Regarding BioID experiments, The authors write “we identified 128 and 60 proteins that significantly interact with CARM1 in HEL and K562 cells.” However, in the figure and the supplementary dataset, I could not find any statistical test that was specifically used to determine this cutoff of interactors vs. non-interactors. This should be clarified, as the

nature of this significance cutoff impacts the downstream network analysis used.

Response: Given space limitations, we did not include the required information. To address this concern, we have now added the descriptions and references in the supplemental methods section.

8. The synergy impacts of the EPZ molecule and ruxolitinib remain relatively unconvincing, at least related to the proposed JAK2-CARM1 axis here. Can the authors provide more evidence that the noted synergy is related to their proposed mechanism? It is easy to imagine this synergy could result from other cellular impacts of these two agents.

Response: To provide clearer interpretation for the readers, we have revised the graphical abstract to explain one possible mechanism about the synergistic effect of inhibiting CARM1 and JAK2 (Figure 7). In addition, aspects of the reviewer's query will need to be more fully answered in future studies, as the questions will require years of inquiry to define the various mechanisms affected by these inhibitors.

Minor

1. In the mass spectrum figure panels (Fig. 1B), it would be preferable to alter the y-axis scale so that the intensity of the noted fragment ions do not overlap the amino acid labels.

Response: We thank the reviewer for pointing this out and have revised the figures as suggested.

2. Based on the mass spectrum shown in Fig S5B, it is not clear whether any fragment ions specifically denote and localize the two proposed methylation sites in this peptide.

Response: To address this concern, we modified the figure, adding the raw data of ion signals in Figure S5A. We now present clearer data showing that the methylated residues in histone 3 were identified in mass spectrometry analysis.

3. Discussion line 460: does the CRISPR screen data in DepMap match that from the DepMap shRNA screens (as referenced in the manuscript), regarding CARM-1 essentiality? In general the CRISPR screen data in DepMap is seen to be more reliable than the shRNA.

Response: We thank the reviewer for pointing this out, and confirmed CARM1 essentiality in both DepMap CRISPR screen and DepMap shRNA screen. We have altered the descriptions in our manuscript to provide a clearer explanation and replaced ref.69 (page 29, line 499-501).

4. Fig. S7D: what type of targeted mass spectrometry approach is used here? Is this parallel reaction monitoring (this appears to be the case based on the Methods)? This approach should be specified in the main text and/or figure legend.

Response: As you pointed out, we used parallel reaction monitoring, and now input the approach in the Figure legend of Figure S7D (which is now Figure S7E), in our revised manuscript.

REVIEWER COMMENTS

Reviewer #1 (Remarks to the Author):

My concerns have been addressed.

Reviewer #2 (Remarks to the Author):

The authors have attempted to address the criticisms raised with the resources at hand improving the manuscript somewhat.

Minor comments

It is still possible the observations esp IP results in HEL are an artefact of amplified JAK2 V617F protein in HEL cells - this weakness should be present in the discussion.

It is concerning that EPO stimulation of WT JAK2 does not result in CARM1 phosphorylation suggesting the pathway is not normally activated under physiological conditions. So what is the physiological role of the pathway? AML is not physiology

Also concerning that JAK2 selective inhibitors do not decrease CARM1 phosphorylation to the same degree suggesting other tyrosine kinases activated by JAK2 V617F are involved (in addition to the contribution by JAK1/TYK2 for JAK activation)

Many protein substrates with tyrosine residues will show phosphorylation if placed in an in vitro kinase assay but no evidence that CARM1 is phosphorylated in vivo by wildtype JAK2. Why should the specificity for protein substrates change between JAK2 wildtype and JAK2 V617F - kinase domain structure is the same. These criticisms should be explained in the discussion.

Reviewer #3 (Remarks to the Author):

Review-Nature Communications (IF 17.69) (Itonaga et al, 2024)

Tyrosine phosphorylation of CARM1 promotes its enzymatic activity and alters its target specificity In this manuscript Itonaga and colleagues show elegant evidence to demonstrate JAK2 mediated tyrosine phosphorylation of CARM1 Y149 and Y334, the functional impact of this phosphorylation and how this contributes to CARM1 nuclear and chromatin localisation. The role of in CARM1 and their functional impact on activity and localisation. The impact of CARM1 phosphorylation on the regulation of RUNX1, and RUNX1-target genes through asymmetrical demethylation is demonstrated, where it can influence haemopoietic cell differentiation and regulation of cell cycle and survival. This study highlights CARM1 as a potential target in addition to JAK2 in JAK2-V617F driven disease, this is important research given the clinical challenges of JAK2 inhibitors.

All criteria are met for a Nature Communications article and the authors have adequately addressed my comments raised in my previous review.

Reviewer #4 (Remarks to the Author):

The author response is thorough and well done, and the group is commended on a very good piece

of work. No further suggestions or comments from this reviewer.

Reviewer #5 (Remarks to the Author):

My concerns have now been addressed upon revision, and I find this manuscript acceptable for publication.

Reviewers #1, 3, 4, and 5 all felt that their concerns were met by our revisions.

Below, we address the minor comments of reviewer #2 and describe how we revised the discussion section of the manuscript.

Minor comments

#1. It is still possible the observations esp IP results in HEL are an artefact of amplified JAK2 V617F protein in HEL cells - this weakness should be present in the discussion.

Response: This is clearly not the case as we detected phospho-CARM1 in patient samples that express JAK2-V617F and we could detect an interaction between JAK2-V617F and CARM1 in SET2 cells and weakly in K562 cells. We do mention the potential “hit and run” mechanism involved in the interaction between JAK2-V617F and CARM1 (page 26, line 450-452.).

#2. It is concerning that EPO stimulation of WT JAK2 does not result in CARM1 phosphorylation suggesting the pathway is not normally activated under physiological conditions. So what is the physiological role of the pathway? AML is not physiology.

Response: We now point out (on the last page of our discussion) that CARM1 phosphorylation may be particularly important to the behavior of JAK2-V617F expressing cells, due to some level of addiction to the mutant kinase. In wild type cells, there may be no addiction; it is even possible that there is a selection against CARM1 phosphorylation due to a possible scaffolding role for unphosphorylated CARM1 in the cytoplasm. There are other possibilities for the observed findings as well. Clearly much more studies need to be done to assess the full spectrum of the biological effects of CARM1 phosphorylation. We do detect phospho-CARM1 in AML cells without mutant JAK2 (Figure 2A), there are other kinases capable of phosphorylating CARM1 in cells. Because normal CD34+ hematopoietic stem cells have lower CARM1 expression than AML cells (Figure 2A and Cancer cell. 2018;33:1111-1127.), we have not yet identified a unique role for phospho-CARM1 in normal hematopoiesis.

#3. Also concerning that JAK2 selective inhibitors do not decrease CARM1 phosphorylation

to the same degree suggesting other tyrosine kinases activated by JAK2 V617F are involved (in addition to the contribution by JAK1/TYK2 for JAK activation).

Response: We, and others, have published on the role of JAK1/JAK2 heterodimers in JAK/STAT signaling, thus we agree that other kinases that either affect the activity of JAK2 or are affected by JAK2, or some related kinases, can be involved in regulating CARM1 phosphorylation and activity (page 27, line 460-462.).

#4. Many protein substrates with tyrosine residues will show phosphorylation if placed in an *in vitro* kinase assay but no evidence that CARM1 is phosphorylated *in vivo* by wildtype JAK2. Why should the specificity for protein substrates change between JAK2 wildtype and JAK2 V617F - kinase domain structure is the same. These criticisms should be explained in the discussion.

Response: Our *in vitro* phosphorylation assays showed that only JAK2 and not JAK1, JAK3, or TYK2 phosphorylated CARM1. Furthermore, we only saw phosphorylation on tyrosine 149 and 334. There are many other tyrosine residues in CARM1 and we have not observed their phosphorylation.

In addition, we do mention that JAK2 protein is auto- and/or cross-phosphorylated in cells with mutant JAK2 but not in cells with wild-type JAK2. This also suggests that phosphorylation of JAK2 is necessary for its ability to phosphorylate CARM1 (page 26, line 450-452).